# Pathway-specific enzymes from bamboo and crop leaves biosynthesize anti-nociceptive C-glycosylated flavones

Yuwei Sun[1,4], Zhuo Chen[1,2,4], Jingya Yang[3,4], Ishmael Mutanda ⬤ [1], Shiyi Li[3], Qian Zhang[1,2], Ying Zhang[1,2], Yulian Zhang[1,2] & Yong Wang[1✉]

C-glycosylated flavones (CGFs) are promising candidates as anti-nociceptive compounds. The leaves of bamboo and related crops in the grass family are a largely unexploited bioresource with a wide array of CGFs. We report here pathway-specific enzymes including C-glycosyltransferases (CGTs) and P450 hydroxylases from cereal crops and bamboo species accumulating abundant CGFs. Mining of CGTs and engineering of P450s that decorate the flavonoid skeleton allowed the production of desired CGFs (with yield of 20–40 mg/L) in an *Escherichia coli* cell factory. We further explored the antinociceptive activity of major CGFs in mice models and identified isoorientin as the most potent, with both neuroanalgesic and anti-inflammatory effects superior to clinical drugs such as rotundine and aspirin. Our discovery of the pain-alleviating flavonoids elicited from bamboo and crop leaves establishes this previously underutilized source, and sheds light on the pathway and pharmacological mechanisms of the compounds.

[1] CAS-Key Laboratory of Synthetic Biology, CAS Center for Excellence in Molecular Plant Sciences, Institute of Plant Physiology and Ecology, Chinese Academy of Sciences, Shanghai 200032, China. [2] University of Chinese Academy of Sciences, Beijing 100039, China. [3] College of Food Science and Technology, Shanghai Ocean University, Shanghai 201306, China. [4]These authors contributed equally: Yuwei Sun, Zhuo Chen, Jingya Yang. ✉email: yongwang@sibs.ac.cn

Plant parts have been used for treatment and management of pain and inflammation since time immemorial. Previous and ongoing work with plant extracts has identified some promising antinociceptive compounds, for example, *C*-glycosylated flavones (CGFs) that have been shown to exhibit analgesic activity in both partially pure extracts[1,2] and purified forms[3,4]. CGFs comprise one or more sugar moieties bound to an aglycone scaffold through a direct C–C glycosidic linkage (Supplementary Fig. 1). Compared to other flavone classes and closely related *O*-glycosylated flavonoids, less is known on *C*-glycosylated flavonoids in terms of their distribution pattern in the plant kingdom, the evolutionary history of pathway-specific genes and their pharmacological mechanism for anti-nociceptive activity and other biological functionalities.

As one of the richest sources of natural flavonoids, plants in the grass family provide a wide array of chemically diverse CGFs in their edible and inedible parts. For the past half a century, immense phytochemical studies on agriculturally important crops, i.e., rice[5,6], wheat[7], maize[8], barley[9], and even cash crops like sugarcane[10,11] have documented and profiled abundant *C*-glycosides in leaves, hulls, and grains, represented by vitexin (Vit), orientin (Ori), schaftoside, carlinoside, and maysin (Supplementary Table 1). Bamboo, although not cultivated for food production, is also one member of the grass family that is widely distributed in the world and has various products useful to humans. Edible parts of bamboo like shoots and seeds are commonly consumed in eastern and southern Asia as a rich dietary source of fiber and nutrients, as well as plant-specific phenolic metabolites[12,13]. However, as is the case with many food crops, the non-edible CGF-rich leaves of bamboo are typically viewed far less valuable and largely abandoned. Even though ancient Chinese did believe that some folk medicine or herbal tea made from bamboo leaves could help remove "toxic heat", resulting in the alleviation of inflammation and pain[14], a poor understanding of both the biosynthetic machinery of CGFs in bamboos and the inadequately defined pharmacological mechanisms severely hindered effective and extensive use of bamboo leaves as analgesics.

The in planta CGF biosynthesis (Fig. 1) has been so far partially established in a few monocotyledons and dicotyledonous plants[15–23]. The pathway is understood to involve upstream precursor biosynthesis and several downstream modification enzymes including cytochrome P450s and glycosyltransferases. After formation by condensation of *p*-coumaroyl-CoA from the phenylalanine pathway and malonyl-CoA from fatty acid biosynthesis, naringenin (Nar), the common starter of all flavonoids, is diverted to hydroxylation on the flavanone scaffold by two P450 oxygenases, flavanone 2-hydroxylases (F2H) and flavanone 3′-hydroxylases (F3′H). While F3′H are generally present in most plants, F2H are highly specific to CGF pathways and the only few cases are reported in cereals[19,20,24]. Subsequently, a *C*-glycosyltransferase (CGT) that acts upon an open-ring form of 2-hydroxylated intermediate is key to *C*-glycoside formation (Fig. 1). CGTs belong to the family of the UDP-dependent glycosyltransferases (UGTs) that utilize activated UDP-sugars as donor. Despite that direct *C*-glycosylation on the final flavonoid skeleton have been documented in *Pueraria lobate*[25], *Gentiana triflora*[26] and *Trollius chinensis*[27], the other existing CGTs exclusively come from UGT708 subfamily and glycosylate 2-hydroxylated intermediates with UDP-glucose (UDP-Glc). Considering the chemical diversity of known CGFs bearing various sugars and different glycosylation patterns other than glucosyl substitution, there is still a huge gap between known CGTs and diverse structures of CGFs. The *C*-glucosylated intermediates are unstable and therefore undergo spontaneous dehydration in acidic solvent (Fig. 1), leading to a mixture of Vit (apigenin 8-*C*-β-D-glucoside)/isovitexin (Isovit, apigenin 6-*C*-β-D-glucoside) and Ori (luteolin 8-*C*-β-D-glucoside)/isoorientin (Isoori, luteolin 6-*C*-β-D-glucoside).

An improved understanding of the biosynthetic pathway of CGFs along with dissection of the pharmacological targets of biological activity can allow more efficient utilization of classically discarded parts of bamboo and crops, as well as hasten sustainable production of CGFs through synthetic biology. In this study, we took advantage of the recent availability of genomes in grass family to exhaustively uncover CGTs and P450 genes involved in CGF biosynthesis. Through comparative genomic and biochemical approaches, bifunctional *C*-arabinosyl/*C*-glucosyltransferring enzymes, a previously unknown branch in CGTs from cereals and bamboos were functionally profiled. We also demonstrated the success of heterologous expression of bamboo-originated proteins by metabolic engineering in an *Escherichia coli* chassis and further evaluated the efficacy of the major CGFs from bamboos in in vivo mouse model.

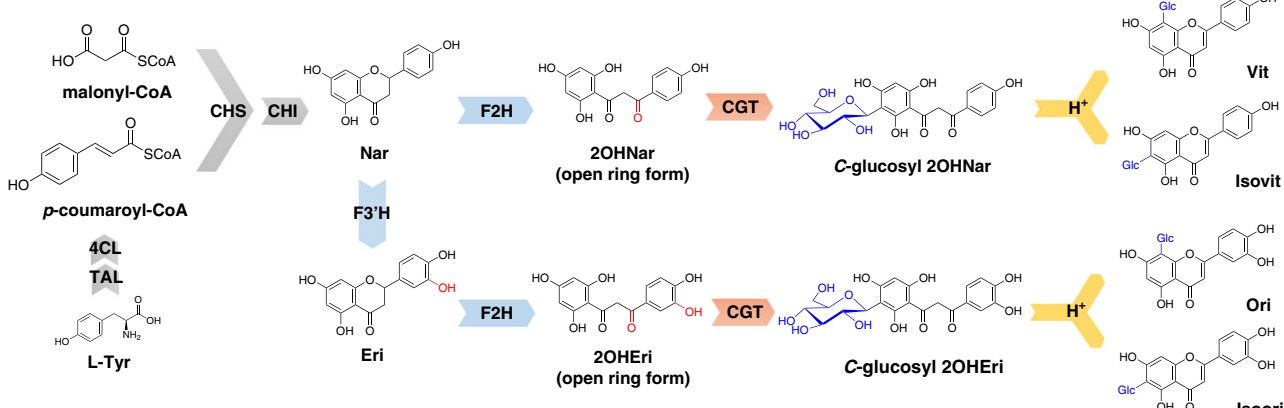

**Fig. 1 Overview of the proposed biosynthetic pathway of *C*-monoglucosylated flavone.** Gray arrows represent the major enzymes responsible for the biosynthesis of naringenin. Blue arrows are flavanone 2-hydroxylases (F2H) and flavanone 3′-hydroxylases (F3′H) decorating the skeleton of flavanone. *C*-glycosyltransferases (CGTs) are indicated in red arrows. After the formation of *C*-glycosylated intermediates, a dehydration reaction occurs spontaneously in acidic solvent (yellow arrows), producing a mixture of 6-*C*- or 8-*C*-glucosides. TAL tyrosine ammonia lyase, 4CL 4-coumarate CoA ligase, CHS chalcone synthase, CHI chalcone isomerase, Nar naringenin, Eri eriodictyol, 2OHNar 2-hydroxylnaringenin, 2OHEri 2-hydroxyleriodictyol, Vit vitexin, Isovit isovitexin, Ori orientin, Isoori isoorientin.

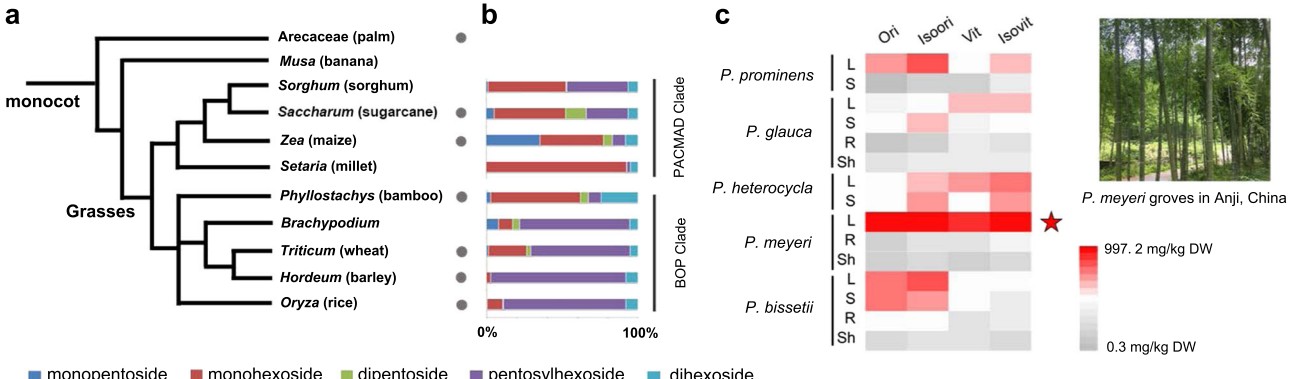

**Fig. 2 C-glycosylated flavone profiling in grass species. a** Phylogenetic tree of monocot species (modified from CoGe). The grass family consists of two major branches: PACMAD clade and BOP clade[29,30]. Gray dots indicate the species that has been reported to accumulate C-glycosylated flavones. Common names are indicated in parentheses. **b** Distribution of C-glycosylated core structures (aglycones bearing only C-sugars) in the Gramineae plants. The proportion of different glycosylated forms are displayed in different colors. High percentage of pentosylhexoside is observed in BOP clade, while PACMAD clade contains more monohexosides. **c** Heat map visualization indicates metabolic spectrum of C-monoglucosylated flavone in different tissues of five *Phyllostachys* bamboos. The contents of major CGF compounds range from 0.3 to 997.2 mg/kg dry weight (DW). Monoglucosides are highly accumulated in the leaves of *P. meyeri* McClure. L leaf, S stem, R root, Sh Shoot.

## Results

**Leaf CGF spectrum of bamboo and related Gramineae crops.**
While CGFs are widely spread among monocot plants (Fig. 2a), especially in the leaves of the grass family[28], a full picture of the CGF diverse spectrum still remains obscure. Here, we applied a liquid chromatography–high resolution-tandem–mass spectrometry (LC–HR–MS/MS) to reveal the profiling of C-glycosylated core structures in the selected Gramineae plants (Fig. 2b). The leaf extracts were treated with hydrochloric acid to remove all O-glycosidic linkages and retain aglycones bearing only C-sugars. Specific $m/z$ values corresponding to different C-glycosylated forms of apigenin, luteolin, and chrysoeriol were extracted and calculated (Supplementary Fig. 2 and Supplementary Table 2), which clearly indicated that PACMAD clade crops (mainly in Panicoideae) accumulate a high proportion of monohexosyl flavones (Fig. 2b). Comparatively, the major glycosylated forms in typical BOP clade plants (rice, wheat, barley, and *Brachypodium*) are diglycosides with both C-pentose and C-hexose moieties, for example, schaftoside, carlinoside, and their isoform from O. sativa[5].

Bamboos (Bambusoideae) are unique non-timber plants that belong to the BOP clade of the grass family[29,30]. Nevertheless, Moso bamboo (*Phyllostachys heterocycla* cv. Pubescens, Synonym: *P. edulis*, Mao-zhu) possesses a C-glycoside profile that resembles those of PACMAD clade (Fig. 2b and Supplementary Fig. 2i). Characteristic fragments in MS/MS spectra also confirmed the presence of various CGF compounds in *P. heterocycle* extracts (Supplementary Fig. 3). Further investigation on the other *Phyllostachys* bamboos showed a diverse pattern of C-glycoside accumulation (Supplementary Fig. 4). Like *P. heterocycla*, *P. meyeri* McClure (a dominant species in Anji county, Zhejiang Province, China, Fig. 2c) also accumulates much more monohexosides than diglycosides, while *P. glauca* McClure, *P. prominens* W. Y. Xiong and *P. bissetii* McClure abundantly produce multiglycosylated forms. The content of major C-monoglucosides (Ori, Isoori, Vit, and Isovit) in different tissues of *Phyllostachys* species is visualized in Fig. 2c. We noticed a tendency that the leaf tissue stores most of the CGFs as expected. Among all the tested samples, *P. meyeri* accumulates 30–60 times higher (iso)vitexin and (iso)orientin than other bamboo species.

**Comparative genomics reveals a rich CGT reservoir.** The chemical diversity of CGFs in the grass family inspired us to explore

the genetic resource responsible for diverse CGF biosynthesis. Due to the close relationship and similar metabolite profiles between Gramineae crops and bamboos, we expected that these plants retained common C-glycoside biosynthesis-relevant genomic segments, which could be traced into their evolutionary history. We first targeted the short arm of rice (*Oryza sativa* japonica) chromosome 6, where the known CGF-biosynthesizing OsCGT[15] (OsUGT708A3) and OsF2H[19] (OsCYP93G2) are located. We carried out comparative genomic analyses and the O. sativa Chr6 was found collinear with those of sorghum (*Sorghum bicolor*, Chr10), foxtail millet (*Setaria italica*, Chr4), *Brachypodium distachyon* (Chr1), bread wheat (*Triticum aestivum*, Chr7A, 7B, and 7D) and maize (*Zea mays*, Chr6, and Chr9) (Fig. 3a). Multiple blocks linked to *T. aestivum* and *Z. mays* can be explained by their recent or ancient polyploidization events[31,32]. The scaffold PH01001494 of Moso bamboo genome[33] can also be partially aligned to these crops (Supplementary Fig. 5).

Discovered in the identified syntenic genome regions of grass family are tandem UGT-coding genes annotated to belong to UGT708 family (Fig. 3b). The rice cultivar Nipponbare (O. sativa japonica) possesses three tandem *UGT708* genes, namely *OsUGT708A2*, *OsUGT708A3*, and *OsUGT708A4*. Their paralogues were observed widely existing in O. sativa indica, S. bicolor, S. italic, B. distachyon, T. aestivum, and Z. mays (Fig. 3b). The number of *UGT708* copies is doubled in the long-grained rice (O. sativa indica) as compared to the japonica variety. By contrast, B. distachyon has only two *UGT708* genes (*BdUGT708A7* and *BdUGT708A8*). We also observed an inversion of the tandem *UGT708* segment on the S. bicolor Chr10 (Fig. 3b). In total, 40 monocot *UGT708* genes (38 uncharacterized with 2 reported) were discovered by this approach (Supplementary Table 3). In order to gain better insights on the evolution of Gramineae UGT708 proteins, we constructed a maximum likelihood (ML) phylogenetic tree using amino acid sequences of the available UGT708 enzymes from both dicot and monocot species (Fig. 3c, for details, see Supplementary Fig. 6). The Gramineae-originated UGT708s form a monophyletic group (UGT708A), establishing a closer relationship with other monocotyledonous UGT708s from orchid, date palm, and yam (Supplementary Table 4). Notably, we found not only clear differentiation of UGT708 proteins between monocot and dicot species, but also the divergence of grass family-specific UGT708A family. UGT708A family is apparently

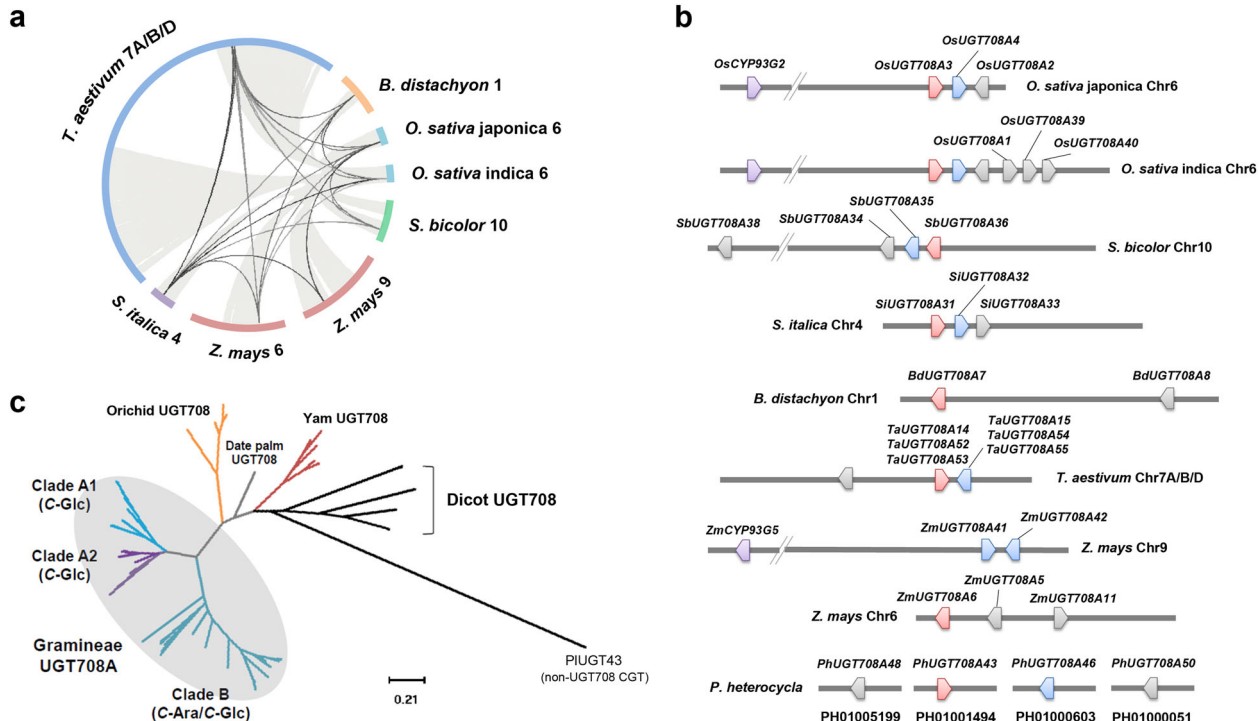

**Fig. 3 Identification of *C*-glycosyltransferases from bamboo-related Gramineae family. a** Syntenic blocks of japonica rice chromosome 6 are found across the genome of Gramineous crops, including *B. distachyon*, *O. sativa* indica, *Z. mays*, *S. bicolor*, *S. italica*, and *T. aestivum*. Locations of CGT-containing regions are linked by black lines. **b** Graphical map of tandem *UGT708A* genes within the syntenic blocks. Clade A1 CGT-encoding genes are indicated in red. Clade A2 CGTs are indicated in light blue. Bifunctional *C*-arabinosyl/glucosyltransferases in Clade B are indicated in gray color. Flavone-2-hydroxylases on the same chromosomes are indicated in purple. For details, see Supplementary Table 3. **c** A Maximum likelihood (ML) phylogenetic tree constructed using PlUGT43 (from *Pueraria lobate*, KU317801.2, as an out-group) and 68 available UGT708 protein sequences. Bar: 0.21 amino acid substitutions per site. Details of the tree are zoomed in Supplementary Fig. 6. *C*-Glc *C*-glucosylation, *C*-Ara *C*-arabinosylation.

separated into Clade A and Clade B, and Clade A can be subdivided into two branches (A1 and A2, Fig. 3c).

**Characterization of *C*-glucosyltransferases and *C*-arabinosyltransferases.** To investigate the putative function of these Gramineae UGT708 enzymes, the open-reading frames of the genes were amplified, cloned, and their proteins expressed. A total of 32 recombinant UGT708s were successfully expressed in *E. coil* (Supplementary Fig. 7) and tested through enzymatic assays. UDP-glucose (UDP-Glc) and UDP-arabinose (UDP-Ara) were used as sugar donors, and phloretin (Phr) and 2-hydroxylnaringenin (2OHNar) were selected as potential acceptors (Fig. 4a).

From the HPLC and LC–MS/MS analyses of the reaction product profiles (Fig. 4b and Supplementary Figs. 8, 9, 11, 12), all enzymes from clade A1 were verified as potent *C*-glucosyltransferases (CGTs) that utilized UDP-Glc to convert both Phr and 2OHNar to their corresponding glycosylated peaks that are identical to the products of positive control (*OsUGT708A3*-mediated reactions) (Fig. 4b, c). No trace of glycosylation was detected in the negative controls (empty vector-mediated reactions). NMR analyses of nothofagin (Nof, Phr 3′-*C*-glucoside) further verified the *C*-glucosylation catalyzed by Clade A1 enzymes (Supplementary Fig. 13). Bamboo CGTs orthologous to PhUGT708A43 were also successfully cloned from *P. meyeri*, *P. glauca*, *P. prominens*, and *P. bissetii*. Multiple variants were found in *P.glauca* and *P. prominens*, while other species seem to hold a conserved CGT completely identical to PhUGT708A43 (Supplementary Fig. 14). The $K_m$ and $k_{cat}/K_m$ values of PhUGT708A43 towards 2OHNar was measured to be 1.75 ± 0.66 µM (Supplementary Table 5) and $4.28 × 10^4\,s^{-1}\,M^{-1}$, which suggested a high affinity and high efficiency. With the exception

of bamboo CGTs and the reported OsUGT708A3, other clade A1 proteins generally favor Phr as indicated by their lower $K_m$ (Supplementary Table 5). On the other hand, most enzymes from Clade A2 just glucosylate Phr to Nof with moderate conversion rate (Fig. 4b). When UDP-Ara was used as sugar donor, only trace of peaks corresponding to *C*-arabinosyl Phr (when 10 times intensified) were observed in the Clade A1/A2-catalyzed reactions (Supplementary Figs. 8b and 9b), which indicated that Clade A is a group of rigid CGTs that highly addict to UDP-Glc.

Subsequently, Clade B were proved to be promiscuous CGTs that consume both UDP-Ara and UDP-Glc, generating the glucosides and arabinosides of Phr and 2OHNar (Fig. 4d, e and Supplementary Figs. 10–13). Most of the Clade B enzymes tolerate both UDP-sugars (Fig. 4a). Interestingly, OsUGT708A39 from *O. sativa* indica predominantly recognizes UDP-Ara, while OsUGT708A2 from *O. sativa* japonica strictly selects UDP-Glc as its donor (Supplementary Fig. 10a, b). OsUGT708A40, ZmUGT708A11, and BdUGT708A8 generated an additional product other than *C*-arabinosyl Phr, which was proposed to be *C*-di-arabinoside of Phr, as indicated in LC–MS/MS experiments (Supplementary Figs. 10e and 11c). Only OsUGT708A1 and OsUGT708A39 were found to slightly arabinosylate 2OHNar detectable in HPLC analyses (Supplementary Fig. 10d).

**Screening of bamboo CYP450 facilitate heterologous *C*-glycoside production in *E. coli*.** In order to achieve a high-yield heterologous *C*-glycoside production platform, we took advantage of the fast-growing and easy manipulation characteristics of *E. coli* system to screen Bamboo F2H and F3′H candidates responsible for *C*-glycoside biosynthesis. The constructed *E. coli* cell factory consisted of a heterologous flavone *C*-glycoside

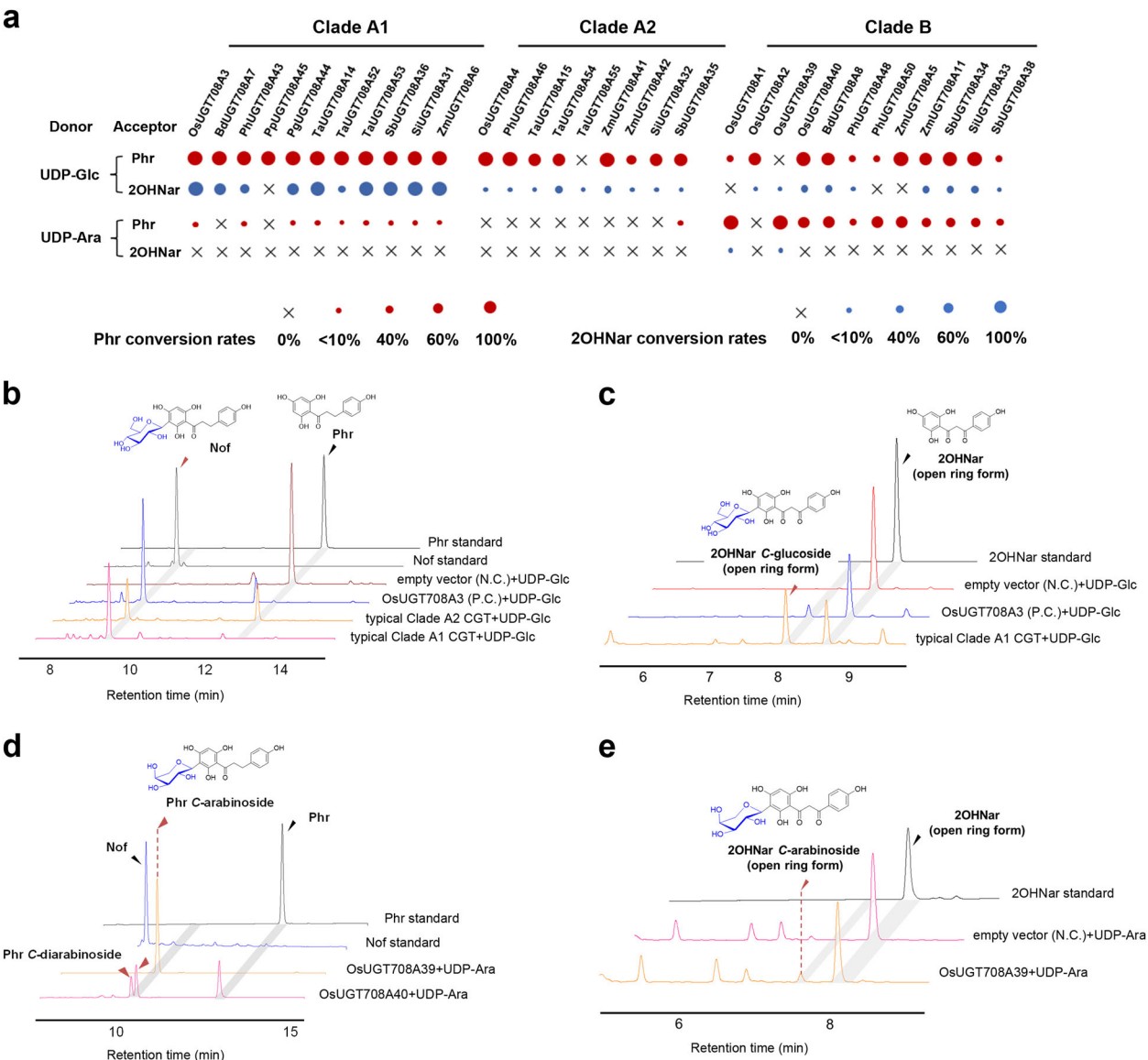

**Fig. 4 Functional characterization of *C*-glucosyl and *C*-arabinosyltransferases. a** Substrate preference of the Gramineae CGTs. The size of the dots indicate proportionally to the conversion rates of Phr and 2OHNar. Clade A enzymes are found to be rigid *C*-glucosyltransferases, while Clade B enzymes are promiscuous to different sugar donors (UDP-Glc/UDP-Ara). HPLC chromatograms of the reactions of Clade A CGTs with UDP-Glc and different receptors including phloretin (Phr, **b**) and 2-hydroxynaringenin (2OHNar, **c**). Clade B CGTs are proved to be able to recognize UDP-Ara, generating *C*-arabinoside of Phr **d** and 2OHNar **e**. The absorption at 280 nm was monitored. For detailed HPLC chromatography of each CGTs (see Supplementary Figs. 8–10; For the MS/MS spectra, see Supplementary Figs. 11, 12).

pathway, which can be split into three modules: the Nar precursor module (Fig. 1, gray arrows), the flavonoid CYP450 decoration module (Fig. 1, navy arrows), and the flavonoid *C*-glycosylation module (Fig. 1, red arrows).

We first utilized a previously assembled pinocembrin-producing strain[34] (sCZ9, Supplementary Table 6) for Nar overproduction. The adopted phenylalanine ammonia lyase (PAL, from *Rhodotorula toruloides*) was proven to tolerate both tyrosine and phenylalanine precursors. To maximize Nar as the precursor of downstream decoration, gradient concentrations of tyrosine were supplemented to boost Nar production to 103 mg/mL (Supplementary Fig. 15). In the next step, three known F2H enzymes from Gramineous crops: the OsF2H (CYP93G2) from rice, the ZmF2H (CYP93G5) from maize, and the SbF2H (CYP93G3) from sorghum were codon optimized, N-terminal modified (transmembrane region truncation and replacement

with 2B1dH-tagged[35]) and tested for their ability to catalyze Nar to 2OHNar in vivo (Fig. 5a). To facilitate the evaluation of validity of the whole pathway, CGT module (PhUGT708A43-expressing plasmid) with confirmative effectiveness was also integrated. When the first three 2B1-infused F2H enzymes coupled with AtCPR2 partner were introduced into *E.coli* respectively, a mixture of Vit and Isovit was detected after a 72 h fermentation (Fig. 5c). An almost complete conversion of Nar was observed from the combination of truncated ZmF2H/AtCPR2 (sCZ2) and truncated SbF2H/AtCPR2 (sCZ29), yielding a highest productivity of 24 mg/L Vit and 27 mg/L Isovit, respectively. Moreover, the accumulation of *C*-glycosyl 2OHNar was also detected in these strains (Fig. 5c). We then attempted to clone F2H-encoding genes from *P. heterocycla* and *P. meyeri* that have not been reported previously. The predicted CDS of *P. heterocycla* CYP93G2-like gene (PH01000832G0680) was redefined through amino acid

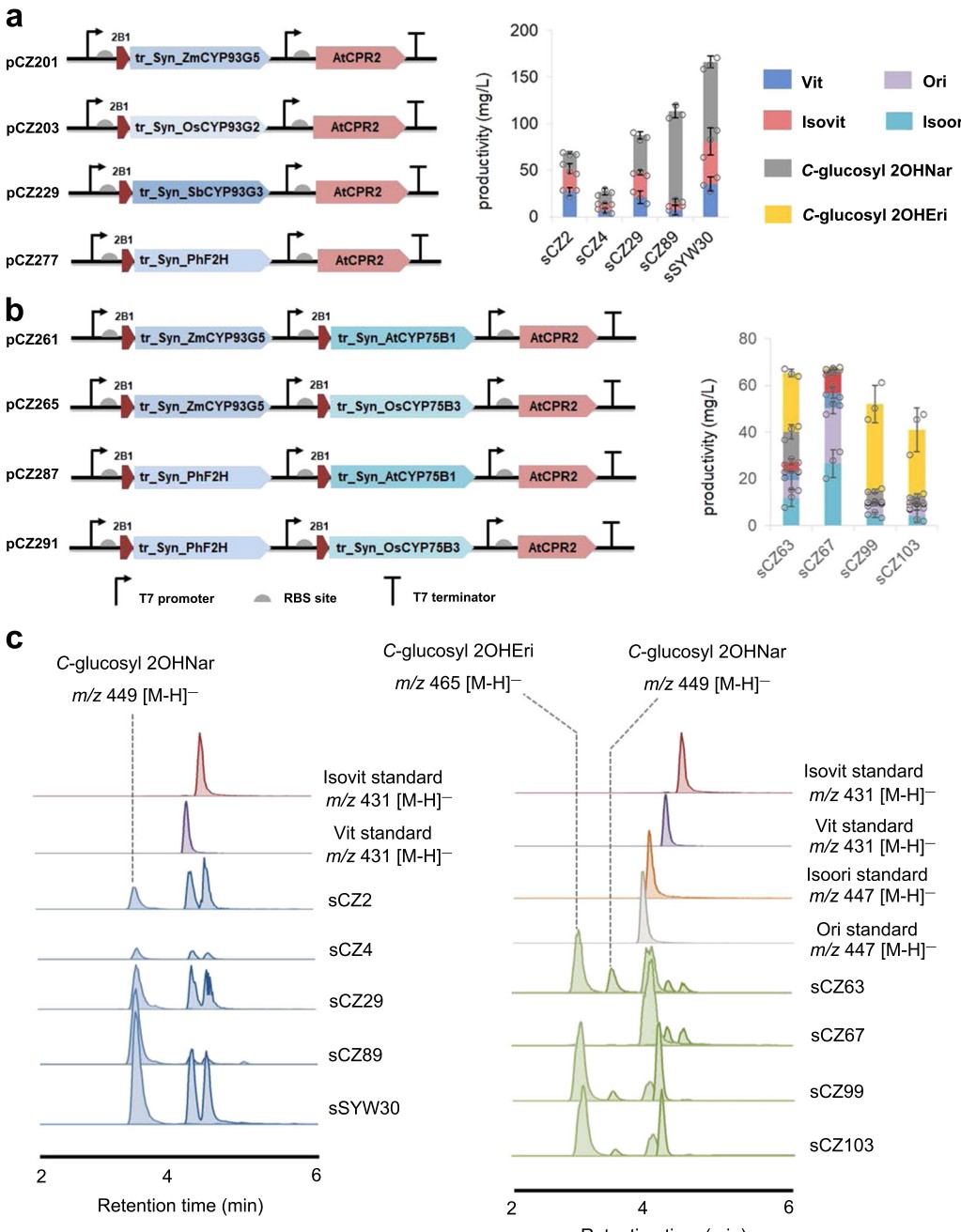

**Fig. 5 De novo biosynthesis of flavone C-glycosides in E. coli cell factory.** The artificial pathway contains tree modules including the naringenin precursor module, the flavonoid CYP450 decoration module, the C-glycosylation module. Details of multispecies genes involved are described in the Supplementary Table 6. **a** Engineered flavone-2-hydroxylases (F2H) from bamboo-related Gramineae family were used to evaluate their ability to produce Vit and Isovit. **b** The complete flavonoid P450 module were further assembled to produce Ori and Isoori. **c** De novo production of flavone C-glycosides. E. coli extracts were sampled 72 or 96 h after induction and analyzed by LC–MS. Target compounds were viewed by ion-extracted chromatogram. Standard samples were also analyzed for comparison.

sequence alignment, since it apparently lacks the N terminal when compared to its homologs in rice and maize. The amplified *CYP93G* genes in *P. heterocycla* and *P. meyeri* (PhF2H/PmF2H) are completely identical and shared 81% identity with the protein sequence of OsCYP93G2. The engineered bamboo CYP93G enzymes showed Vit/Isovit productivity comparable to that of OsCYP93G2, but also accumulated much more *C*-glycosyl 2OHNar than those of other strains, indicating a great potential of *C*-glycoside production. When feeding Nar directly into the strain harboring both PhF2H and PhUGT708A43 (sSYW30), Vit and Isovit increased to $35.1 \pm 7.5$ and $45.8 \pm 14.9$ mg/L (Fig. 5a).

For the optimization of F3′H, the codon-optimized *CYP75B* family genes from different species, namely *CYP75B1* from *Arabidopsis thaliana*, *CYP75B2* from *Petunia* X *hybrida*, *CYP75B3* and *CYP75B4* from rice (*O. sativa*) were N-terminal modified, assembled with CPR partners and introduced into *E. coli* to evaluate their production of eriodictyol. Fermentation of strains sCZ47, sCZ49, and sCZ51 (Supplementary Table 6) led to the accumulation of 30–69 mg/L eriodictyol, while CYP75B4-expressing sCZ53 failed to catalyze a 3′-hydroxylation reaction (Supplementary Fig. 16). Four *F3′H*-like genes were obtained from *P. meyeri*, which are identical to the predicted sequences of

*PH01000052G1210*, *PH01000926G0660*, *PH01002485G0110*, and *PH01001730G0080* in *P. heterocycla*. Among them, only the protein product of *PH01001730G0080* (Ph/PmF3′H) exhibit comparable 3′-hydroxylase activity toward Nar with 40.7% conversion rate (Supplementary Fig. 16).

We finally constructed the complete flavonoid P450 module by inserting the most active F3′H-encoding genes (*AtCYP75B1* and *OsCYP75B3*) between F2H (ZmCYP93G5 or PhF2H) and AtCPR2 (Fig. 5b), and integrated the constructs into *E. coli* with a Nar module and CGT module. The resulting strains sCZ63, sCZ67, sCZ99, and sCZ103 (Supplementary Table 6) produced a mixture of four flavone *C*-glycosides. From simple precursors glucose and tyrosine, strain sCZ67 (ZmCYP93G2-OsCYP75B3-CPR) produced 26.6 mg/L Isoori and 23.9 mg/L Ori without accumulation of *C*-glucosylated flavone intermediates (Fig. 5c). The final products were proved by large-scale fermentation, isolation, and NMR-based identification (Supplementary Fig. 17).

**Isoorientin exhibits both neuro-analgesic and anti-inflammatory activities**. To preliminarily screen the antinociceptive effect of *C*-glycosides, the four major monoglucosides (Ori, Isoori, Vit, and Isovit) were assessed in acetic acid-writhing, hot plate and formalin-induced (first and second phase) nociception assays using ICR male mice. All the four compounds showed significant analgesic activity in acetic acid writhing and formalin-induced nociception tests, but no effects on the latency time on the hot plate (Supplementary Fig. 18). Generally, isoorientin (Isoori) exhibited the best effects among the tested compounds. Isoori (7.5–30 mg/kg) exerted analgesic effect in a dose-dependent manner, and 30 mg/kg Isoori inhibited acetic acid-induced writhing by 86.2%, licking time by 73.9% in the first phase and 48.3% in the second phase of formalin-induced nociception (Fig. 6a). The anti-nociception action of Isoori was equivalent to that of the positive reference drug rotundine (ROT) in the first phase and acetylsalicylic acid (ASA) in the second phase.

Since isoorientin was observed to be effective towards both the first and second phases of formalin-induced nociception, it was considered as a promising agent with both neuro-analgesic and anti-inflammatory activities. Next, we investigated its potential molecular targets through evaluating the effects on some of the neurotransmitter or physical/chemical stimuli-activating receptors. We selected the GluN2B subunit of N-methyl-D-aspartate receptors (NMDARs)[36] and transient receptor potential vanilloid 1 (TRPV1)[37] due to their important roles in pain transmission and modulation. Isoori (30 mg/kg) significantly suppressed TRPV1 protein expression in the spinal dorsal horn of mice and decreased the level of GluN2B in a dose-dependent manner (Fig. 6b). On the other hand, the anti-inflammatory effects of Isoori were indicated by its inhibition of TNF-α, IL-1β, and promotion of anti-hyperalgesic cytokine IL-10 production in formalin-induced mice paw inflammation (Fig. 6c).

Given that Isoori belongs to flavonoids that have been reported to have antioxidant activity[38,39], its effects on the oxidative stress indicators like glutathione synthetase (GSH), malondialdehyde (MDA), superoxide dismutase (SOD), and ascorbic acid (VC) levels were also investigated. Compared to the vehicle solution, Isoori effectively increased the concentrations of GSH, SOD, and VC in paws injected with formalin, but suppressed formalin stimulated MDA production in a dose-dependent manner (Supplementary Fig. 19)

We also evaluated potential toxicity of Isoori in the same mice model through oral administration of a high dose in an acute toxicity test. After 14 days of a single administration (5000 mg/kg), 4 of the 14 mice were dead, and the living mice did not show any behavioral alterations. The survival curve (Supplementary Fig. 20) shows that the $LD_{50}$ of Isoori on the ICR mice is more than 5000 mg/kg, which conforms to a low toxicity profile.

**Discussion**

In light of the rising opioid crisis, development of natural analgesic drugs has been receiving increasing interest. Plants and microbes have provided humankind with some of the most clinically relevant drugs and secondary metabolites with beneficial effects on various ailments including pain relief. For example, some recent successful cases reported are indole alkaloids from *Nauclea latifolia*[40,41] and *Corydalis yanhusuo*[42], and diterpene glucosides from *Rhododendron micranthum*[43]. Flavonoids are an important class of phytochemicals ubiquitously existing in the plant kingdom and have members that are considered promising candidates for natural analgesics[44]. To enable full utilization of flavonoids, there is a strong need to uncover more bioresources rich in flavones, and to improve our understanding of the biosynthetic machinery and mechanisms by which flavonoids confer their beneficial effects. Our present work revealed bamboo species as a rich source of CGFs, and elucidation of the biosynthetic pathway uncovered *C*-glucosyl/*C*-arabinosyl-transferring UGT708A enzymes. We further demonstrate the construction of the pathway in a heterologous host, and show that major CGFs in bamboo, especially isoorientin, has both neuroanalgesic and anti-inflammatory effects superior to clinical drugs used as positive controls.

Plants have evolved two pathways for glycosylating the flavone scaffold through *O*-glycosylation and *C*-glycosylation. Unlike *O*-glycosylation that terminates flavone biosynthesis, *C*-glycosylation occurs on unstable intermediates prior to the final scaffold formation. Except for PiUGT43[25], GtUF6CGT1[26], and TcCGT1[27], all characterized CGTs to date belong to the UGT708 family[15–18,22,23], indicating that the UGT708 family may be specialized for *C*-glycosylation of phenylpropanoids and flavonoids. Through this study, we have significantly expanded the knowledge on monocot UGT708A family by not only tripling the number of known CGTs from current 12 (2 monocot, 7 dicot UGT708, and 3 non-UGT708 CGTs) to 38, but also by revealing a group of hitherto unknown group that could recognize UDP-Ara (Figs. 3, 4 and Supplementary Table 3). *C*-arabinosides are unique to the leaves of grass family plants. Arabinosyltransferases of plant origin have been reported to be involved in the arabinoxylan biosynthesis[45–47] and in post-translational modification of glycoproteins[48], but UDP-arabinosyltransferases decorating secondary metabolites are hardly reported with just two cases[49,50] (identified as *O*-arabinosyltransferases). Recent genetic-based studies on rice and maize metabolism have implied some gene candidates for *C*-arabinosylation[51–54], but none have been functionally characterized to date. The Clade B CGTs found in this research are promiscuous *C*-arabinosyl/glucosyltransferases, which is considered as a group of specialized CGTs evolving for more diverse recognition of sugar donors. The grass family plant may take the advantage of these potent Clade B CGTs to diversify their *C*-glycoside spectrum, producing abundant *C*-arabinosylated flavone compounds.

Screening of P450 oxidases in this research revealed bamboo flavanone 2-hydroxylases (Ph/PmF2H) and flavanone 3′-hydroxylases (Ph/PmF3′H). The F2Hs from bamboo and crops are CGF pathway-specific enzymes that result in the formation of unstable chalconoid intermediates readily dehydrated to flavones[19]. In our in vivo *E. coli*-based cell factory, instead of 2-hydroxylated compounds, a great accumulation of *C*-glycosylated 2-hydroxynarigenin or 2-hydroxyeriodictyol was observed (up to 97 mg/L, Fig. 5). This phenomenon was not mentioned in the

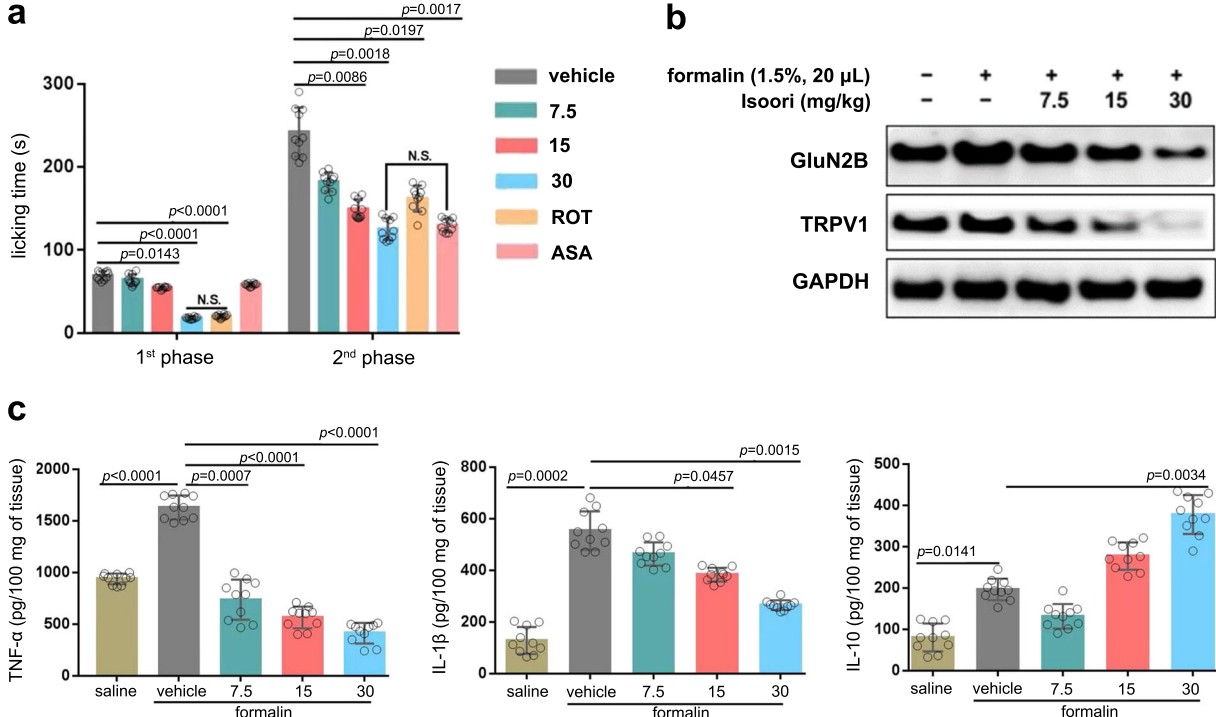

**Fig. 6 Antinociceptive effects of isoorientin. a** Effects of Isoori on the first and second phases of formalin-induced nociception in mice were tested in different concentrations (7.5, 15, 30 mg/kg, i.p). Reference drug [acetylsalicylic acid (ASA, 200 mg/kg, p.o.), rotundine (ROT, 20 mg/kg, i.p.)] was administered to mice. Details are described in "Methods" section. Each column represents mean ± SEM ($n = 10$). $P < 0.05$ was considered as statistically significant. **b** Isoori suppressed the synaptic accumulation of NMDA receptor GluN2B subunits and TRPV1 expression levels in the spinal dorsal horn of mice. The level of GluN2B and TRPV1 were checked using immunoblotting analysis with GAPDH as internal reference. Experiments were repeated three times with similar results. **c** Effects of Isoori (7.5, 15, 30 mg/kg) on TNF-α (left), IL-1β (mid), and IL-10 (right) production in formalin paw inflammation. Drug administration were described in "Methods" section. TNF-α tumor necrosis factor-α, IL-1β interleukin-1β, IL-10 interleukin-10.

prior reports of metabolic engineering on CGF biosynthesis in yeast systems[55,56], which indicated that *E. coli* probably lacked endogenous dehydratases active on 2-hydroxylated flavanone. The successful bio-production of (iso)orientin and (iso)vitexin here was achieved through synthetic approach, which involves multiple bioparts from bamboo, maize, rice, and mouse-ear cress (*A. thaliana*). The ratio of 6-*C*- and 8-*C*- products is generally 6:4 that likely reflect the result of a non-enzymatic reaction.

The present study also demonstrates that the four CGFs dominant in leaves of bamboos and crops showed significant antinociceptive and anti-inflammatory effects (Fig. 6). Isoorientin exhibited the best analgesic activity in both the first and second phase of the formalin assay, displaying anti-nociceptive activity equivalent to the reference drug ROT, a dopamine receptor antagonist in the first phase, and ASA, an anti-inflammatory agent in the second phase (Fig. 6a). This was interesting, given that first phase pain responses are mainly neurogenic pain response[57], while second phase pain responses are mediated through inflammatory mechanisms[58]. Opioid analgesics are known to exert their anti-nociception effects on phase I and phase II, while non-steroidal anti-inflammatory drugs (NSAIDs) only inhibit phase II pain alleviation. A previous study showed that Vit exhibits protection against isoflurane-induced increase in expression of GluN2B and TRPV1 in PC12 cells[59]. In the present work, in vivo evidence of suppression of TRPV1 and NR2B by isoorientin in the spinal dorsal horn was provided. Compared to aforementioned examples and the mechanisms of most clinical compounds, Isoori is herein reported as a highly effective compound to exhibit both analgesic and anti-inflammatory activity which are equivalent to the clinical commonly used drugs, respectively.

Taken together, this study revealed the normally underutilized leaves of bamboos and other cereal crops as a rich source of CGFs. We leveraged the availability of sequenced plant genomes to exhaustively search CGTs and shed more light on the metabolic diversity of flavonoids in plants. In addition to the evolutionary relationships of CGTs unraveled here, this work tripled the number of known CGTs by successful characterization of specific CGTs with glucosyl and arabinosyl-transferring activities. Crucial insights in the biosynthesis of CGF in bamboos are presented, together with a demonstration of the potential to utilize these availed genomic resources in metabolic engineering in *E. coli*. Finally, we provide insights on the anti-nociceptive and anti-inflammatory activity of isoorientin in in vivo models.

## Methods

**Extraction and profiling of flavone *C*-glycosides**. The frozen leaf samples (100 mg) were crushed and extracted overnight at room temperature with 5 mL 80% methanol. The crude extracts were filtered, concentrated in vacuo to 0.5 mL aqueous solution and treated with 1 M hydrochloric acid (HCl) in boiling water for 6 h. The acid-treated samples were extracted three times with *n*-butanol. The combined butanol extracts were further evaporated to dryness and re-dissolved in 1 mL 50% HPLC grade methanol before LC–MS/MS analysis.

Ultra-performance liquid chromatography (UPLC)-electrospray ionization (ESI)-high resolution (HR) tandem mass spectrometry (MS/MS) data were acquired using Q Exactive hybrid quadrupole-Orbitrap mass spectrometer (Thermo Scientific, USA) equipped with an Acquity UPLC BEH C18 column (∅ 2.1 × 50 mm, 1.7 μM, Waters, USA). The mobile phase consisted of methanol (0.1% formic acid, solvent A) and $H_2O$ (solvent B). A linear gradient was set as follows: 0.0–10.0 min, 5–100% A in B; 10.0–12.5 min, 100% A; 12.5–15.0, re-equilibrate to the initial condition. The flow was 0.25 mL/min. The mass acquisition was performed in negative ionization mode with full scan (50–1000). The results are shown in Fig. 2 and Supplementary Fig. 2.

**Bioinformatics analysis**. Genomes of the grass family plants were downloaded from Ensembl Plants. Multiple collinearity of subgenomes was detected by MCScanx[60] with default parameters and plotted by circle plotter (Fig. 3a). The Gramineae *UGT708*-encoding genes were identified by Plant Compara within Ensembl Plants. Genome or transcriptome of banana (*Musa acuminata*), date palm (*Phoenix dactylifera*), yam (three *Dioscorea* species), and orchid (*Phalaenopsis equestris*) (Supplementary Table 4) were BLASTed for *UGT708*-encoding genes. Sequences longer than 300 amino acids were selected and subjected into phylogenetic tree construction, along with Gramineae UGT708A and dicot UGT708 enzymes. The phylogenetic trees of UGT708 proteins were generated by Mega X using ML method with bootstrap analyses of 1000 replications (Fig. 3c and Supplementary Fig. 6). The multiple alignments of amino acid sequences were generated by using Clustal Omega program provided by the European Molecular Biology Institute.

**In vitro enzymatic assay and kinetic studies of CGTs**. A total of 32 recombinant CGTs were successfully cloned using gene-specific primers (Supplementary Table 7) and expressed in *E. coli* (see Supplementary Method). A typical enzymatic assay was performed in a 100 μL aliquot of reaction mixture containing buffer A (100 mM NaCl, 20 mM Tris–HCl, pH 8.0), 400 μM UDP-glucose or 200 μM UDP-Ara, 100 μM chalconoid acceptors [Phr, 2-hydroxynaringenin (2OHNar)], 25 μL crude (or 5 μg purified) enzymes. The reaction mixtures were incubated at 37 °C for 2 h and finally quenched by 100 μL methanol and vortexed vigorously. The kinetic parameters of CGTs were determined in a 50 μL reaction solution consisting of buffer A, 1–10 μg each purified enzyme and 1–100 μM Phr and 2OHNar. The concentration of UDP-glucose was fixed at 400 μM. The reactions were incubated for 20 min at 37 °C and stopped by the addition of 50 μL methanol.

The quenched reaction mixtures were centrifuged (12,000 rpm) for 15 min and subjected to HPLC analyses (Dionex UltiMate 3000SD HPLC system, Thermo Scientific, USA). Separation was achieved on a C18 column [SilGreen ODS column (∅ 4.6 × 250 mm, S-5 μM), Greenherbs Co., Ltd., Beijing, China] with a flowrate of 1 mL/min at 40 °C. Detection was performed with mobile phases containing acetonitrile (0.1% formic acid, solvent A) and $H_2O$ (0.1% formic acid, solvent B) under a linear gradient elution: 0–20 min, 5–100% A in B, 100% A maintained for 5 min. The injection volume was 20 μL and absorption was monitored at $\lambda = 280$ nm (Fig. 4). The enzyme kinetic parameters were calculated by quantification of the formation of glucoside products. Michaelis–Menten curves were generated by GraphPad Prism 5.

**Cloning of flavonoid hydroxylases, construction of heterologous CGF synthetic pathway, and de novo production**. The candidate F2Hs from various plants (OsCYP93G2 XM_015787468.2, SbCYP93G3 XM_002461241.2, ZmCYP93G5 XM008661918.2, and Ph/PmF2H) and F3′Hs (AtCYP75B1 AF271651.1, PxhCYP75B2 AF155332.1, OsCYP75B3 XM_015757555.2, OsCYP75B4 XM_015757228.2, and Ph/PmF3′H) were codon-optimized and synthesized by GenScript Biotech (Nanjing, China). Their N-terminal transmembrane helices were predicted through TMHMM Server v.2.0, truncated and replaced with 2B1dH tag. The engineered F2Hs, F3′Hs were assembled with AtCPR2 (NM_179141.2) into pETDuet-1 by use of ClonExpress II One Step Cloning Kit (Vazyme, Nanjing, China) (Supplementary Table 6, for primers see Supplementary Table 7). For the construction of dual flavonoid hydroxylases cassette, the engineered AtCYP75B1 and OsCYP75B3 were, respectively, inserted into *BamH*I/*Not*I-double digested pCZ201 and pCZ277 through one-step cloning, resulting in pCZ261, pCZ265, pCZ287, and pCZ291 (Supplementary Table 6).

The plasmid pYH055[34] with spectinomycin resistance was transformed into the *E. coli* BL21(DE3) (designed as strain sCZ9) for the Nar production. The plasmid pCZ86 (pET28a-PhUGT708A43) with kanamycin resistance was chosen for the CGT module. For the de novo production of CGFs from tyrosine, the assembled flavonoid hydroxylases and CPR cassettes were cotransferred with pYH055 and pCZ86 into *E. coli* BL21(DE3). The seeds were precultured at 37 °C in LB medium overnight and then inoculated (1:100) into MOPS minimal medium supplemented with 5 g/L glucose and 2.5 g/L L-tyrosine. After the OD$_{600}$ reached to 1.0, IPTG (0.1 mM) was added to the cultures. Subsequently, the cultures were incubated at 22 °C and 220 rpm for 3–4 days. Aliquots of the cultures (500 μL) were taken every day for monitoring the target products and subsequently supplemented with 0.5% glucose. The samples were extracted by 500 μL *n*-butanol for three times. The combined supernatant was evaporated under vacuum and dissolved in 100 μL methanol; 2 μL was injected for UPLC–MS/MS analysis (Fig. 5c). Condition of LC–MS/MS was identical to that described above.

For the production of CGFs from Nar, only bamboo P450 (pCZ277) and CGT (pCZ86) module were incorporated to give strain sSYW30. Nar (25 mg/L) was supplemented into the medium for every 24 h for four times. The extraction and chromatographic analyses were performed in the way as the samples of de novo production from tyrosine.

**Antinociceptive assay**. The acetic acid-induced test was carried out according to the method described previously[61]. Ori, Vit, Isovit (30 mg/kg, i.p.) and isoorientin (7.5, 15, 30 mg/kg, i.p.) prepared above or vehicle (5% DMSO in normal saline, i.p.) was administered to mice 1 h before intraperitoneal (i.p.) injection of acetic acid

(0.6% in distilled water, 10 mL/kg). ASA (200 mg/ kg, p.o.) was employed as reference drug, which was dissolved in the same vehicle. The number of writhes was counted for 15 min. The hot-plate test measured response latencies according to the method described by Eddy and Leimbach[62]. Animals were placed on an PE34 (IITC, Life Science Instrument, USA) hot-plate maintained at 55 ± 1 °C and the time between placement of the animal on the hot-plate and the occurrence of either the licking of the hind paws, shaking, or jump off from the surface was recorded as response latency. Mice with baseline latencies of <5 s or more than 30 s were eliminated from the study 24 h previously. Animals were treated with different compounds (Ori, Vit, Isovit, 30 mg/kg; isoorientin, 7.5, 15, 30 mg/kg, i.p.) or morphine (5 mg/kg, i.p.) 1 h before the experiments. Control animals received the same treatment as mentioned in the abdominal writing test. The formalin test was similar to that described previously[58]. Twenty microlitres of 1.5% formalin were injected subcutaneously into the right hind paw of mice. The time (in seconds) spent licking and biting the injected paw was taken as an indicator of nociception response. Responses were measured for 0–10 min (first phase) and 10–60 min (second phase) after formalin injection, respectively. Ori, Vit, Isovit (30 mg/kg, i.p), isoorientin (7.5, 15, and 30 mg/kg, i.p.), ROT (20 mg/kg, i.p.), or ASA (200 mg/ kg, p.o.) was administered 1 h before formalin injection and control animals received the same volume of vehicle (0.5% DMSO in distilled water). All experiments were approved by the Institutional Animal Care and Use Committee (IACUC) of Shanghai Ocean University and were performed in accordance with national and institutional guidelines for the care and use of laboratory animals.

**Western blot**. In another cohort of animals with identical treatment, at 1 h after the injection of formalin when the pain sensitivity reached its peak, both ipsilateral and contralateral dorsal horns of lumbar spinal cord (from segments L3 to L5) were dissected after a laminectomy under deep anesthesia of formalin. The dorsal horn sample was homogenized in ice-cold Laemmli buffer, containing 50 mM Tris–HCl, pH 7.5, 0.5% sodium dodecyl sulfate (SDS), 5% 2-mercaptoethanol, and 1% protease inhibitor cocktail (Millipore Sigma, St. Louis, MO, USA). The proteins were separated on SDS–polyacrylamide gel electrophoresis (SDS–PAGE) and then transferred to polyvinylidene difluoride membranes. Membranes were then incubated with different primary antibodies including GluN2B (Cell Signaling Technology, Danvers, USA) with a 1:1000 dilution, GAPDH (Proteintech, USA) with a 1:5000 dilution, Trpv1 (Abcam, England) using a 1:500 dilution overnight at 4 °C. Membranes were then incubated with goat anti-rabbit IgG (Beyotime Biotechnology, China) or goat anti-mouse IgG using a 1:1000 dilution for 1 h at room temperature. Proteins were detected by BCA protein quantification kit (Sangon biotech., Shanghai, China) and visualized on X-ray films. Densitometry was analyzed by ImageJ software (National Institutes of Health, Bethesda, MD, USA). The density of specific bands of Western blot was then normalized to corresponding loading control bands.

**Reporting summary**. Further information on research design is available in the Nature Research Reporting Summary linked to this article.

## Data availability

The gene sequences of *Phyllostachys* bamboos were deposited in GenBank under the following accession numbers: PhUGT708A43 (PhCGT1), MK616588; PhUGT708A46 (PhCGT2), MK616589; PhUGT708A48 (PhCGT3), MK616590; PhUGT708A50 (PhCGT4), MK616591; PhUGT708A44 (PgCGT1), MK616592; PpUGT708A45 (PpCGT1), MK616593; PhF2H, MK628906; PhF3′H, MK636710. Any other data not present in the manuscript or supplementary materials is available from the authors upon reasonable request.

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

## Acknowledgements

This work was financially supported by the National Key R&D Program of China (2018YFA0900600), the Strategic Priority Research Program "Molecular mechanism of Plant Growth and Development" of CAS (XDB27020202, XDB27020103), the National Natural Science Foundation of China (Grant nos. 31670099, 31700261). This work was also financially supported by the Construction of the Registry and Database of Bioparts for Synthetic Biology of the Chinese Academy of Science (No. ZSYS-016), the International Partnership Program of Chinese Academy of Science (No. 153D31KYSB20170121) and the National Key Laboratory of Plant Molecular Genetics, SIPPE, CAS. We would like to thank Dr. Wenjuan Yuan, Wenzhi Zhou, Yuanhong Shan, and Shizhen Bu in the Core Facility Centre of SIPPE for technical assistance on high-resolution MS and NMR acquisition. We thank Dr. Lei Yang (Shanghai Chenshan Plant Science Research Center, CAS) for providing the *Phyllostachys* bamboo materials.

## Author contributions

Y.W., Y.S., and J.Y. conceived the study. Y.S. performed the bioinformatics analysis, biochemical experiments, and prepared the manuscript. Z.C. constructed the plasmids, strains, and carried out microbiological manipulations and extractions. J.Y. and S.L. accomplished the pharmacological study. Y.S., I.M., J.Y., and Y.W. wrote the paper. Q.Z., Ying Z. and Yulian Z. contributed to enzyme kinetic study and data analysis. All authors contributed to discussion of the manuscript.

## Competing interests

The authors declare no competing interests.
