## [Peer Review File · Communications Biology]

Reviewers' comments:

Reviewer #1 (Remarks to the Author):

This paper investigated bamboo and cereal crop leaves as a rich source of C-glycosylated flavones. Novel pathway-specific enzymes including C-glycosyl transferases (CGTs) and P450 hydroxylases and the potential to utilize genomic resources in metabolic engineering in E.coli were reported, and give promising candidates as anti-nociceptive compounds. The work is meaningful and the conclusions are original and of interest for the scientists in this areas.

Some minor revisions should be considered.

Line 585-586, Orientin, vitexin, isovitexin (30 mg/kg, i.p) and isoorientin (7.5, 15, 30 mg/kg) or vehicle (5% DMSO in normal saline) was administered to mice 1 hour before intraperitoneal (i.p.) injection of acetic acid (0.6% in distilled water, 10 mL/kg). please add the administering manner of isoorientin and vehicle. Are they also i.p.?

For the Plant materials, please give some identification information, including specimen and voucher No.

Line 462-463, isoorientin is herein reported as the first compound to effectively exhibit both analgesic and anti-inflammatory activity. Please make the statement more Strictness and objectivity, and based on more evidence.

Reviewer #2 (Remarks to the Author):

The authors of "Bamboo and Crop Leaves Biosynthesize Antinociceptive C-glycosylated Flavones" reported C-glycosyltransferases (CGTs) and P450 hydroxylases from cereal crops and bamboo species. Various approaches were applied to screen and analyze the candidate genes of CGTs. The pathway of C-glycosylated flavones (CGFs) was rebuilt in the E. coli cell factory. And the target CGFs isoorientin and orientin were successfully produced with the yield of 20–30 mg/L. Although the data are rich and solid, the article was not well organized and written and the comments are as follows.

1. The title of the article is inappropriate and cannot cover the main content.
2. Monocot UGT708A genes were divided into Clade A and Clade B in the maximum likelihood phylogenetic tree. Clade A was for glucosyltransferases and Clade B was for arabinosyltransferases. First, the phylogenetic tree was not reliable as the monocot CGTs were from very limited plants. Most of them were from the same plant and exhibited very high homology. Second, how were the CGTs of Clade B assigned as arabinosyltransferases? Is it possible that these CGTs possess catalytic promiscuity in recognizing UDP-Ara? If not, please present the evidence such as Km of CGTs while different sugar donors were used. Third, for the whole article, what's the purpose of this part "Evolution of CGT confers chemical diversity of C-glycosides"? This part should be divided into two parts ("Finding" and "Analyzing"). The authors should also provide the SDS-PAGE of all the expressed 34 recombinant UGT708s in the SI.
3. The identification of representative CGFs should be clear. Please provide the NMR data of the final products generating by the cell factory. Additionally, the products of C-arabinosyltransferases should also be isolated and identified.
4. Is the yield of the CGFs obtained by the standard curve? Please make it clear.
5. The part of discussion was redundant and some of the points have been mentioned in the results.
6. Line 63-66, Page 4. Please correct the statement and add the ref (He J B, et al. Angewandte Chemie, 2019.)
7. Line 81, Page 5. The statement of "6C-/8C-..." was confused and wrong. Line 120 "colors"
8. Line 314, Page 14. "mg" should be "mg/L"

9. Some figures are not well drawn and very confused. For example, Fig5B the marks around the AtCPR2 were different. Different promoters? Fig 3E, what's the meaning of the CGTs in grey? Fig 3F what did the two red triangles represent? Please make it clear in the Legend.

10. There are also many small mistakes in the references (ref 4, 11, 14, 17, 18, 58, 65, 69 et al). Please check it carefully.

Reviewer #3 (Remarks to the Author):

This manuscript mainly describes the gene discovery and metabolic engineering involved in the biosynthesis of C-glycosylated flavones (CGFs), as well as the pharmacological studies on the antinociceptive activity of some CGF compounds. The manuscript was well organized and the work was evidenced by a lot of data including many aspects of the secondary metabolism and functional genomic researches on CGFs in grass family plants. With regard to the bioactivity part, the authors reported isoorientin as a unique antinociceptive compounds with both neuroanalgesic and anti-inflammatory effects.

This manuscript perfectly combined evolutionary, synthetic and pharmacological biologies together to study the natural products. It is worth publishing on this journal after revision.

Major comments:

(1) I was confused by the pathway in Fig1. Why the nature open the ring and then close the ring again. Please clarify this issue.

(2) It would be better to combine the results from Fig3 and Fig4. The major point is the evolution of novel function for CGT gene family.

Minor comments:

(1) Page 6. In Figure 2C, the scale bar beside the heatmap seems indicates the MS signal intensity, can authors quantify the CGF amount in different bamboos using absolute values? Since the authors had the standards, it would be better to provide the absolute quantification to compare the CGF accumulation in bamboo with other plants

(2) In the second result, the authors described more than 30 C-glycosyltransferases (some have already been systematically named) from the grass family, I suggest the authors to name these genes/proteins through UGT Nomenclature Committee. Current numbering of these enzymes like OsCGT1, 2, 3 is confusing.

(3) Page 11. Line 235. "In order to chieve a high-yield", I think it's a mistake.

(4) In the results of heterologous production of CGFs in E coli systems, the authors did not mentioned the reason of choosing this host. Previous work in yeast (García Vanegas K, Microbial Cell factory (2018) 17: 107) reported comparative yield of these compounds, which raised question of advantage of adopting E coli as alternative host.

(5) Figure 5B, in the illustration of cz261, the arrow and bar are misplaced.

Response to Reviewer Comments (COMMSBIO-19-0957A)

We thank the reviewers for their helpful comments and valuable suggestions for improving our work and manuscript. The comments are very useful in helping us to strengthen the results, especially for the characterization of novel *C*-arabinoxyltransferases. We have carefully revised the manuscript in accordance with the reviewers' comments. All the modifications in the manuscript and SI are indicated in red font. The response to the reviewers are listed point by point as follows:

Reviewer #1 (Remarks to the Author):

This paper investigated bamboo and cereal crop leaves as a rich source of *C*-glycosylated flavones. Novel pathway-specific enzymes including *C*-glycosyl transferases (CGTs) and P450 hydroxylases and the potential to utilize genomic resources in metabolic engineering in *E.coli* were reported, and give promising candidates as anti-nociceptive compounds. The work is meaningful and the conclusions are original and of interest for the scientists in this areas.

Some minor revisions should be considered.

“Line 585-586, Orientin, vitexin, isovitexin (30 mg/kg, i.p) and isoorientin (7.5, 15, 30 mg/kg) or vehicle (5% DMSO in normal saline) was administered to mice 1 hour before intraperitoneal (i.p.) injection of acetic acid (0.6% in distilled water, 10 mL/kg). Please add the administering manner of isoorientin and vehicle. Are they also i.p.?”

Response: We thank the reviewer for pointing out this issue. They are also administered i.p. We have revised this mistake in the manuscript.

“For the Plant materials, please give some identification information, including specimen and voucher No.”

Response: Thanks for the suggestion. We have added the information of plant materials in SI Supplementary Methods. We provided the voucher specimen No. of *Phyllostachys* bamboos and cultivar information of each Gramineae plants.

Line 462-463, isoorientin is herein reported as the first compound to effectively exhibit both analgesic and anti-inflammatory activity. Please make the statement more Strictness and objectivity, and based on more evidence.

Response: Thanks for the suggestion. In this study, we have investigated the nociceptive effect of major CGF compounds (especially isoorientin) through formalin test in mice. Opioid like morphine, codeine and oxycodone are active in both early phase (phase I, neuro-analgesic) and late phase (phase II, anti-inflammatory) [1,2]. For non-opioid compounds, they are usually phase-specific analgesics. However, after checking the reference we have found some recent cases show both phase I and phase II inhibition, which is quite like isoorientin. DHCB (dehydrocorybulbine, from *Corydalis yanhusuo*) (10~40 mg/kg) has been reported as antinociceptive in both inflammatory and neuropathic pain models [3]. DHCB also exhibits weak activity at the μ receptor, and therefore cannot be excluded from opioid antagonist; chrysin, a flavone compound, in a dosage of 150 mg/kg shows notable activity in both phases in formalin test [4]. Although there are not many cases, “isoorientin is the first compound effectively exhibit both analgesic and anti-inflammatory activity” was overstated. We apologize for this mistake and modified the sentence with more objectivity. We have replaced the sentence with “isoorientin is herein reported as a highly effective compound to exhibit both analgesic and anti-inflammatory activity which are equivalent to the clinical commonly used drugs respectively.”

Reviewer #2 (Remarks to the Author):

The authors of “Bamboo and Crop Leaves Biosynthesize Antinociceptive C-glycosylated Flavones” reported C-glycosyltransferases (CGTs) and P450 hydroxylases from cereal crops and bamboo species. Various approaches were applied to screen and analyze the candidate genes of CGTs. The pathway of C-glycosylated flavones (CGFs) was rebuilt in the *E. coli* cell factory. And the target CGFs isoorientin and orientin were successfully produced with the yield of 20–30 mg/L. Although the data are rich and solid, the article was not well organized and written and the comments are as follows.

1. The title of the article is inappropriate and cannot cover the main content.

Response: We agree with the reviewer. We have revised the title of the article to “Pathway-specific Enzymes from Bamboo and Crop Leaves Biosynthesize Antinociceptive C-glycosylated Flavones”

2. Monocot UGT708A genes were divided into Clade A and Clade B in the maximum likelihood phylogenetic tree. Clade A was for glucosyltransferases and Clade B was for arabinosyltransferases. First, the phylogenetic tree was not reliable as the monocot CGTs were

from very limited plants. Most of them were from the same plant and exhibited very high homology. Second, how were the CGTs of Clade B assigned as arabinosyltransferases? Is it possible that these CGTs possess catalytic promiscuity in recognizing UDP-Ara? If not, please present the evidence such as Km of CGTs while different sugar donors were used. Third, for the whole article, what's the purpose of this part "Evolution of CGT confers chemical diversity of C-glycosides"? This part should be divided into two parts ("Finding" and "Analyzing"). The authors should also provide the SDS-PAGE of all the expressed 34 recombinant UGT708s in the SI.

Response: Thanks for the constructive suggestions from the reviewer. In this comment, the reviewer asked four questions:

(I) For the first question, we agree with the reviewer that the "monocot CGTs" in the previous tree just cover limited species from monocot plants (only CGTs from the grass family were included). To clarify the phylogenetic relationship of UGT708s and reconstruct a more reliable tree, we introduced some non-grass UGT708s from other monocotyledon plants listed in **Table S3** in SI. Genome or transcriptome of banana (*Musa acuminata*), date palm (*Phoenix dactylifera*), yam (three *Dioscorea* species) and orchid (*Phalaenopsis equestris*) were BLASTed for *UGT708*-encoding genes. Most of the obtained UGT708 sequences were fragment probably due to the low assembly quality caused by their complex genomes. Sequences longer than 300 amino acids were selected and subjected into phylogenetic tree construction.

Origin	Common Name	Order	Family	Sequence Assembly	UGT708	ID	Len (AA)	Type
Musa acuminata	banana	Zingiberales	Musaceae	ASM31385v1 (Genome)	0	/	/	/
Phoenix dactylifera	date palm	Arecales	Arecaceae	PDK30 (v3) (Genome)	2	PDK_30s886991g001	265	fragment
						PDK_30s888691g001	309	fragment
Dioscorea composita	Barbasco	Dioscoreales	Dioscoreaceae	GBJW01.1 (Transcriptome)	6	GBJW01072389.1	444	full-length
						GBJW01072383.1	444	full-length
						GBJW01072385.1	211	fragment
						GBJW01072384.1	198	fragment
						GBJW01019112.1	168	fragment
Dioscorea rotundata	white Guinea yam	Dioscoreales	Dioscoreaceae	TDr96_F1_Pseu do_Chromosome_v1.0 (Genome)	5	Dr03079.1.cds	468	full-length
						Dr03081.1.cds	460	full-length
						Dr03078.1.cds	359	fragment
						Dr03080.1.cds	277	fragment
						Dr03077.1.cds	776	fragment
Dioscorea zingiberensis	peltate yam	Dioscoreales	Dioscoreaceae	GBCR01.1 (Transcriptome)	2	GBCR01066044.1	363	fragment
						GBCR01038049.1	162	fragment
Phalaenopsis equestris	/	Asparagales	Orchidaceae	OrchidBase 3.0 (Genome)	5	PEQU_16406	427	fragment
						PEQU_16402	386	fragment
						PEQU_16404	370	fragment
						PEQU_37780	324	fragment
						PEQU_40915	68	fragment

Table S3 in SI. Non-grass family monocotyledonous UGT708 identified from the genome or transcriptome of banana, date palm, yam and orchids. No *UGT708* genes was found in banana

(*Musa acuminata*). Date palm, yam and orchid possess 2~6 *UGT708*-encoding genes, though most of these genes are not in full length.

From the re-constructed maximum likelihood (ML) tree [**Figure 3C** (a concise version) in the manuscript, **Figure S7** (a detailed version) in SI], we can conclude that **(I)** *UGT708*s from the grass family still form a monophyletic group (*UGT708A*). This is also supported by the systematic naming of *UGT708A* by UGT Nomenclature Committee (as requested by Reviewer #3): All the grass family CGTs found in this study were defined as members of *UGT708A* subfamily (see response to Reviewer #3, and also the updated **Table S2** in SI). **(II)** *UGT708*s from date palm, yam and orchid are closer to *UGT708A* than to the dicot *UGT708*s, while the subfamilies of these genes are yet to be identified. It is not correct to define all monocot CGTs as *UGT708A* (date palm, yam and orchid *UGT708*s are not on the same branch of the Gramineae-specific *UGT708A*). Based on the new information, we modified **Figure 3C** in the manuscript by introducing more monocot *UGT708*s, moving the zoom-in Clade A and B trees to the SI to make **Figure 3** more concise, and also we revised the description of “Monocot *UGT708A*” to “Gramineae *UGT708A*”.

Figure 3C and S7. Phylogenetic tree of UGT708 family. PIUGT43 is used as an out-group. **(A)** **Figure 3C** in the manuscript, a concise version without gene names; bar: 0.21 amino acid substitutions per site. The function of the Gramineae CGTs was defined as *C-Glc* (*C*-glucosylation) and *C-Ara* (*C*-arabinsylation). **(B)** **Figure S7**, a detailed version. Bootstrap values (based on 1000

replications) are indicated at each node. Dicot UGT708 enzymes include MiCGT (KT200208.1) [5], FeUGT708C1 (BAP90360.1), FeUGT708C2 (BAP90361.1) [6], GmUGT708D1 (LC003312.1) [7], FcUGT708G1 (LC131333.1), CuUGT708G2 (LC131334.1) and ChUGT708G3 (LC131335.1) [8]. Gramineae-specific UGT708A genes (**Table S2** in SI) are divided into Clade A1, A2 and Clade B. Information of the other monocotyledonous UGT708s from orchid, date palm and yam are listed in **Table S3** in SI.

(II). In the second question, the reviewer mentioned the possibility of Clade B (*C*-arabinosyltransferases) being promiscuous to multiple UDP-sugar donors. It is reported that CGTs in dicot plants can recognize multiple UDP-sugar donors, for example, MiCGT (accepting UDP-glucose/xylose) [5]; FeUGT708C1/C2 (accepting UDP-glucose/xylose) [6]; FcUGT708G1/CuUGT708G2 (accepting UDP-glucose/xylose/galactose) [8]. Some of the “direct” CGTs (directly *C*-glycosylate the flavone/isoflavone skeleton, and they do not belong to the UGT708 family) are also reported to be promiscuous to different UDP-sugars. GtUF6CGT1 tolerates both UDP-glucose and UDP-galactose [9], and TcCGT1 accepts UDP-glucose/xylose/arabinose [10]. Nevertheless, UDP-glucose are always identified as the most effective donors in these *C*-glycosyltransferases.

In our case, we carefully tested all the Gramineae UGT708A enzymes by use of UDP-glucose (UDP-Glc) and UDP-arabinose (UDP-Ara) as sugar donors. Phloretin (Phr) and 2-hydroxylnarigenin (2OHNar) were chosen as sugar acceptors. The conversion rates of enzymatic reactions are summarized in **Figure 4A**. In **Figure 4A**, the sizes of the dots proportionally indicate the conversion rates. The HPLC analyses of each enzyme (tested in four combinations: UDP-Glc/Phr, UDP-Ara/Phr, UDP-Glc/2OHNar, UDP-Ara/2OHNar) are shown in **Figure S9-11** in SI.

When UDP-Glc was used as sugar donor, Clade A1 enzymes efficiently convert Phr and 2OHNar to their *C*-glucoside (**Figure S9A** and **S9C**) (nothofagin was supported by NMR and LC-MS/MS data; *C*-glucosyl 2OHNar was supported by LC-MS/MS). The glucosylation efficiency of Clade A2 are relatively low (**Figure S10A** and **S10C**), which is probably due to the poor expression of these proteins. Both Clade A1 and A2 enzymes barely accept UDP-Ara as sugar donors, only when phloretin were used as sugar acceptor, trace activities toward UDP-Ara (detectable if the signals were 10 times intensified) were observed in nine of eleven Clade A1 enzymes, as well as

SbUGT708A35 in Clade A2 (**Figure S9B** and **S10B**).

In the previous manuscript, we defined Clade B as a group of *C*-arabinosyltransferases. After checking the promiscuity of sugar donors (**Figure S11**), we found that Clade B is actually a group of bifunctional *C*-arabinosyl/glucosyltransferases. Most of the Clade B enzymes tolerate both UDP-sugars (**Figure 4A**). Interestingly, OsUGT708A39 from *O. sativa* indica predominantly recognizes UDP-Ara, while OsUGT708A2 from *O. sativa* japonica strictly selects UDP-Glc as its donor (**Figure S11A** and **S11B**). ZmUGT708A11, OsUGT708A40 and BdUGT708A8 generated an additional product other than *C*-arabinosyl Phr, indicating that these CGTs could be di-arabinosyltransferases (**Figure S11E**).

Overall, we conclude that Clade A (A1/A2) CGTs in the grass family are rigid *C*-glucosyltransferases and Clade B are promiscuous *C*-arabinosyltransferases/glucosyltransferases. We consider that Clade B is a group of specialized CGTs evolving for more diverse recognition of sugar donors. In dicot plants, diversified *C*-glycosides may be resulted from the promiscuity of CGTs which mainly produce *C*-glucosides, but also *C*-xylosides/galactosides/arabinosides as side products (usually present in relatively low amount). However, in the grass family, some potent *C*-mono/diarabinosyltransferases in Clade B like OsUGT708A1, OsUGT708A39, OsUGT708A40, BdUGT708A8, PhUGT708A50, ZmUGT708A5, ZmUGT708A11, SbUGT708A34, SiUGT708A33 confer potent arabinoside-producing abilities. The Grass family plant may take the advantage of Clade B to diversify their *C*-glycoside spectrum, which is highly characteristic of *C*-arabinosides.

We also added the kinetic parameters of some CGTs in Clade A2 and Clade B (**Table S4** in SI). The K_m value of some CGTs were not calculated, which is because of the low conversion rates. The highly costly UDP-arabinose is another obstacle to measure to the kinetic parameter with UDP-ara.

Figure 4A. Substrate preference of UGT708A enzymes. The size of the dots are proportional to the conversion rates. Clade A enzymes are found to be rigid C-glucosyltransferases, while Clade B enzymes are more promiscuous to different sugar donors (UDP-Glc/Ara).

Figure S9. HPLC analyses of the enzymatic reaction mixtures of Clade A1 CGTs. Eleven Clade A1 CGTs were tested towards 4 combinations of substrates: **(A)** Phr/UDP-Glc; **(B)** Phr/UDP-Ara; **(C)** 2OHNar/UDP-Glc; **(D)** 2OHNar/UDP-Ara. The Clade A1 enzymes barely generate arabinosides, but efficiently catalyze the glucosylation of Phr and 2OHNar.

Figure S10. HPLC analyses of the enzymatic reaction mixtures of Clade A2 CGTs. Clade A2 CGTs were tested towards 4 combinations of substrates: **(A)** Phr/UDP-Glc; **(B)** Phr/UDP-Ara; **(C)** 2OHNar/UDP-Glc; **(D)** 2OHNar/UDP-Ara. Clade A2 enzymes showed moderate conversion of Phr to nothofagin (Nof).

Figure S11. HPLC analyses of the enzymatic reaction mixtures of Clade B CGTs.

Clade B CGTs were tested towards 4 combinations of substrates: **(A)** Phr/UDP-Glc; **(B)** Phr/UDP-Ara; **(C)** 2OHNar/UDP-Glc; **(D)** 2OHNar/UDP-Ara. Clade B enzymes recognize both UDP-Glc and UDP-Ara as sugar donors. The red triangles in **(D)** indicate the proposed C-arabinosyl 2OHNar. **(E)** HPLC analyses revealed the presence of proposed phloretin C-diarabinoside (indicated by red triangles).

Table S4. K_m values of recombinant CGTs toward 2-hydroxynaringenin and phloretin.

Data are presented as means \pm S.D. from triplicate measurements ($n = 3$).

^a K_m values were measured with UDP-glucose (400 μ M) as sugar donor.

^b Previously reported data¹⁶.

^c cannot be determined because no detectable conversion was observed.

K_m (μ M) ^a	UDP-Glc (as sugar donor)	
	phloretin	2-hydroxynaringenin
OsUGT708A3	4.78 ^b	2.5 ^b
BdUGT708A7	2.72 \pm 1.39	24.27 \pm 10.37
PhUGT708A43	15.83 \pm 6.98	1.75 \pm 0.66
PpUGT708A45	12.73 \pm 4.69	/ ^c
PgUGT708A44	12.87 \pm 4.46	2.75 \pm 0.75
TaUGT708A14	4.57 \pm 2.19	25.82 \pm 6.62
TaUGT708A52	7.71 \pm 3.42	/ ^c
TaUGT708A53	19.01 \pm 4.07	19.40 \pm 7.55
SbUGT708A36	4.22 \pm 2.48	28.01 \pm 7.85
SiUGT708A31	5.32 \pm 0.80	19.29 \pm 5.42
ZmUGT708A6	11.47 \pm 2.48	48.31 \pm 18.26
OsUGT708A4	3.47 \pm 1.73	/ ^c
PhUGT708A46	76.35 \pm 41.16	/ ^c
TaUGT708A15	26.54 \pm 3.33	/ ^c
TaUGT708A54	44.12 \pm 8.37	/ ^c
TaUGT708A55	/ ^c	/ ^c
ZmUGT708A41	39.13 \pm 12.29	/ ^c
ZmUGT708A42	15.73 \pm 6.71	/ ^c
SiUGT708A32	22.00 \pm 9.24	/ ^c
SbUGT708A35	27.05 \pm 11.59	/ ^c
OsUGT708A1	/ ^c	/ ^c
OsUGT708A2	51.06 \pm 30.64	/ ^c
OsUGT708A39	/ ^c	/ ^c
OsUGT708A40	116.8 \pm 33.7	/ ^c
BdUGT708A8	17.97 \pm 3.94	/ ^c
PhUGT708A48	/ ^c	/ ^c
PhUGT708A50	/ ^c	/ ^c
ZmUGT708A5	36.92 \pm 7.87	/ ^c
ZmUGT708A11	35.66 \pm 8.29	/ ^c
SbUGT708A34	19.15 \pm 4.92	/ ^c
SiUGT708A33	21.06 \pm 8.50	/ ^c
SbUGT708A38	/ ^c	/ ^c

(III). For the third question, in **Result 1** we have investigated the C-glycoside spectrum of the grass family plants and noticed that they are abundant in both mono- and di-hexoside or pentosides, so in **Result 2** “**Evolution of CGT confers chemical diversity of C-glycosides**” we would like to explain how the grass family plants use their specified CGTs to produce diverse C-glycosides with hexosyl (glucosyl) or pentosyl (arabinosyl) moieties. The connectivity of these two parts are not clear, which makes it hard to follow the logic. We added one sentence: “*The chemical diversity of CGFs in the grass family inspired us to explore the genetic resource responsible for diverse CGF-biosynthesis.*” to explain our motivation. We agree with the suggestion that this part (**Result 2**) should be divided into two parts: one emphasizing gene discovery (“Finding”) and another being focused on functional analysis (“Analyzing”). According to the comments from the reviewer, we have modified the manuscript and split **Result 2** into two sub-sections which are entitled as “**2. Comparative Genomics Reveals a Rich C-glycosyltransferase Reservoir**” and “**3. Characterization of C-glucosyltransferases and C-arabinosyltransferases**”.

We now consider that “Evolution of the CGTs” in this manuscript is not well explained, and also for the whole article, the evolution part contribute little to the understanding of CGT function. Therefore, we have removed the some redundant description of CGT divergence to make the results and discussion part more concise.

(IV). Thanks for the kind reminder. We have added the SDS-PAGE and western bolt analysis of 32 recombinant UGT708s in **Figure S8** in SI. We corrected the number “34” to “32” since the nucleotide sequence of PmCGT1 (from *P. meyeri*) and PbCGT1 (from *P. bissetii*) were found to be completely identical to PhUGT708A43 (PhCGT1, from *P. heterocycla*). From the SDS-PAGE results (**Figure S8A** and **S8C**), we noticed that most of the Clade A1 (rigid C-glucosyltransferases) and Clade B (promiscuous C-arabinosyl/glucosyltransferases) were expressed when using *E. coli* BL21(DE3) as expression host, except for SbUGT708A38. The soluble expression of Clade A1 and Clade B proteins greatly facilitated the glycosylation of chalcone substrates (almost 100% conversion of Phr by Clade A1, see Figure R3;). Clade A2 were not well expressed and could only be detected by western bolt (**Figure S8B**). Nevertheless, the crude enzymes extracts were also observed to be active towards Phr and 2OHNar (See **Figure S10**)

Figure S8. Expression of recombinant UGT708A enzymes in *E. coli*.

All the His6-tagged proteins were purified by nickel magnetic beads. **(A)** SDS-PAGE indicated that Clade A1 enzymes (except for PpUGT708A45) were well expressed in *E. coli* BL21(DE3). **(B)** The expression of Clade A2 could only be detected by western blot. Although Clade A2 were not well expressed, the crude enzymes extracts were observed to be active towards Phr and 2OHNar (See **Figure S10**). **(C)** SDS-PAGE of Clade B enzymes indicated that most of them were expressed.

3. The identification of representative CGFs should be clear. Please provide the NMR data of the final products generating by the cell factory. Additionally, the products of *C*-arabinosyltransferases should also be isolated and identified.

Response: We agreed with the reviewer. We have provided the ^1H NMR spectra of four final products (orientin, Ori; isoorientin, Isoori; vitexin, Vit; isovitexin, Isovit) produced by the cell factory in SI (**Figure S18**). The products were isolated from 1 L fermentation of strain SCZ67 (for *C*-glycoside production profiling, see **Figure R1**). The extraction and isolation procedures is described in SI Supplementary Methods.

We also provided the NMR spectra of two *C*-glycosides of phloretin (**Figure S14** in SI). 3'-*C*-glucosyl phloretin (nothofagin, Nof) was proved by ^1H , ^{13}C NMR and 2D NMR (HMBC). The ^1H NMR spectrum of *C*-arabinosyl phloretin is no so clear, probably due to the low amount of the product (2 mg UDP-arabinose was reacted with phloretin in a 56% conversion rate), or probably due to the instability (no natural phloretin arabinoside was reported). The isolation is described in SI Supplementary Methods.

Figure R1. An extract fraction containing *C*-glucosyl flavones and *p*-coumaric acid.

Figure S18. NMR analyses of flavone C-glucosides produced by *E. coli*.

(A) ^1H NMR (500 MHz, $\text{DMSO-}d_6$) spectrum of the purified orientin (Ori, luteolin 8-C-glucoside); (B) ^1H NMR (500 MHz, $\text{DMSO-}d_6$) spectrum of the purified isoorientin (Isoori, luteolin 6-C-glucoside); (C) ^1H NMR (500 MHz, $\text{DMSO-}d_6$) spectrum of the purified isovitexin (Isovit, apigenin 6-C-glucoside); (D) ^1H NMR (500 MHz, $\text{DMSO-}d_6$) spectrum of the purified vitexin (Vit, apigenin 8-C-glucoside).

Figure S14. NMR spectra of nothofagin and phloretin C-arabinoside.

(A) ^1H NMR (500 MHz, $\text{DMSO-}d_6$) spectrum of nothofagin (Nof, phloretin 3'-C-glucoside); (B) ^{13}C NMR (125 MHz, $\text{DMSO-}d_6$) spectrum of Nof; (C) HMBC spectrum of Nof reveal a direct C-C linkage indicated by long-rang correlation from anomeric proton to C2' and C3'; (D) ^1H NMR (500 MHz, $\text{DMSO-}d_6$) spectrum of proposed phloretin C-arabinoside.

4. Is the yield of the CGFs obtained by the standard curve? Please make it clear.

Response: The yield of CGF compounds was quantified by the standard curves showed in **Figures**

R2:

Figure R2. Standard curves of C-glycosylated flavones generated by HPLC.

The mobile phase consisting of methanol (0.1% formic acid) and H₂O (0.1% formic acid) can well distinguish vitexin (Vit) and isovitexin (Isovit), but is unable to achieve baseline separation of orientin (Ori) and isoorientin (Isoori) (**Figure R3A**). In contrast, a acetonitrile (0.1% formic acid)-H₂O (0.1% formic acid) system well separates Ori and Isoori, but not Vit/Isovit isomers (**Figure R3B**). Therefore each sample for the quantification of C-glucosylated flavones was run twice on the same HPLC system using methanol-H₂O or acetonitrile-H₂O as respective mobile phase. The yields were calculated based on the peak area of well-separated peaks.

Figure R3. Separation of four major C-glucosylated flavones by different HPLC conditions.

(A) Separation was achieved on C18 ODS column using methanol [a linear gradient elution of 5% to 100% (20 min)] in H₂O (flow rate of 1 mL/min, detected at 280 nm).

(B) Separation was achieved on C18 ODS column using acetonitrile [a linear gradient elution of 10% to 50% (20 min)] in H₂O (flow rate of 1 mL/min, detected at 280 nm).

5. The part of discussion was redundant and some of the points have been mentioned in the results.

Response: Thanks for the suggestion. We have shortened the discussion part by deleted some redundant points. To make the whole article more concise, we have also removed some points about the evolution and also, discussion of Morphi-Dex development.

6. Line 63-66, Page 4. Please correct the statement and add the ref (He J B, et al. Angewandte Chemie, 2019.)

Response: In this part, we mentioned potential dehydratase that may catalyze the elimination of the 2-hydroxyl group. It is only hypothesized in rice and wheat [11], however, has not been proved yet. In dicot plants, in particular in those “direct” C-glucosylation case (like in *Pueraria lobate* [12],

Gentiana triflora [9] and *Trollius chinensis* [10]), CGTs specifically add C-glucosyl group on the C-6 or C-8 position of flavone/isoflavone skeletons. No dehydration is needed. We consider it is not necessary to mention dehydration in the pathway, therefore we revised this part, removed the description of yet-to-be proved dehydratase and also revised **Figure 1**. The recent reference of TcCGT1 from *Trollius chinensis* [10] was added in Line 65, Page 4, being grouped with other direct C-glucosylation examples.

7. Line 81, Page 5. The statement of “6C-/8C-...” was confused and wrong. Line 120 “colors”

Response: Thanks for the suggestion. “6C-/8C-” refers to the position of C-C linkage between flavone skeleton and sugar. We have correct the confusing description of “6C-/8C-”. 6-C- and 8-C- isomers were replace by specific compound names: vitexin (Vit, apigenin 8-C- β -D-glucoside), isovitexin (Isovit, apigenin 6-C- β -D-glucoside), orientin (Ori, luteolin 8-C- β -D-glucoside) and isoorientin (Isoori, luteolin 6-C- β -D-glucoside). We also revised Line 120 “color” to “colors”.

8. Line 314, Page 14. “mg” should be “mg/L”

Response: Thanks for pointing out this mistake. We have corrected this mistake in the manuscript.

9. Some figures are not well drawn and very confused. For example, Fig5B the marks around the AtCPR2 were different. Different promoters? Fig 3E, what’s the meaning of the CGTs in grey? Fig 3F what did the two red triangles represent? Please make it clear in the Legend.

Response: We apologize for the mistakes in the figures. We have revised the marks in **Figure 5B**. The promoters are the same as those of other genes. In the previous **Figure 3E**, the CGTs indicated in grey is the UGT708 paralogs that (1) not collinear with other Gramineae-specific CGTs (For example, tandem CGTs of *Triticum aestivum* are proposed to locate on Chr. 7, but some other UGT708-encoding genes are also found on Chr. 1) or (2) those genes were failed to be cloned. We have moved **Figure 3D** and **3E** to **Figure S7** in SI, these genes in the tree are now showed in a compressed branch. In the previous **Figure 3F**, two red triangles indicated the divergence time of Clade A/Clade B enzymes and Clade A1/Clade A2 enzymes. Since we decided not to overstate the parts about evolution, we deleted **Figure 3F** in the manuscript.

Figure 5. *De novo* biosynthesis of flavone C-glycosides in *E. coli* cell factory.

The artificial pathway contains three modules including the naringenin precursor module, the flavonoid CYP450 decoration module, the C-glycosylation module. Details of multispecies genes involved are described in the Table S5. (A) Engineered flavone-2-hydroxylases (F2H) from bamboo-related Gramineae family were used to evaluate their ability to produce vitexin and its derivatives. (B) The complete flavonoid P450 module were further assembled to produce orientin

and isoorientin. (C) *De novo* production of flavone C-glycosides. *E. coli* extracts were sampled 72 or 96 h after induction and analyzed by LC-MS. Target compounds were viewed by ion-extracted chromatogram. Standard samples were also analyzed for comparison.

10. There are also many small mistakes in the references (ref 4, 11, 14, 17, 18, 58, 65, 69 et al). Please check it carefully.

Response: Thanks for the suggestion. We have carefully check the reference and revised the mistakes in the manuscript and SI.

Reviewer #3 (Remarks to the Author):

This manuscript mainly describes the gene discovery and metabolic engineering involved in the biosynthesis of C-glycosylated flavones (CGFs), as well as the pharmacological studies on the antinociceptive activity of some CGF compounds. The manuscript was well organized and the work was evidenced by a lot of data including many aspects of the secondary metabolism and functional genomic researches on CGFs in grass family plants. With regard to the bioactivity part, the authors reported isoorientin as a unique antinociceptive compounds with both neuroanalgesic and anti-inflammatory effects.

This manuscript perfectly combined evolutionary, synthetic and pharmacological biologies together to study the natural products. It is worth publishing on this journal after revision.

Major comments:

(1) I was confused by the pathway in Fig1. Why the nature open the ring and then close the ring again. Please clarify this issue.

Response: It is really interesting that the C ring of flavanone skeleton is once formed and then opened again by F2H (flavanone 2-hydroxylase) for C-glycosylation decoration. The C ring is thought to be opened by CYP93G (in monocots) [13,14] and CYP93B (in dicots) [15,16] monooxygenases when 2-hydroxyflavanone compounds are tautomerized to dibenzoylmethane-

form intermediates (**Figure R4**). The dibenzoylmethane intermediates are highly unstable and readily to undergo spontaneous dehydration, which results in more stable flavone skeleton. This is one of three pathways to form the flavone skeleton [(**I**) FNS I-mediated, (**II**) FNS II-mediated, (**III**) F2H+dehydration-mediated pathway]. Unlike the direct FNS I/FNS II pathway, the F2H pathway looks inefficient because two steps are needed (**Figure R4**). However, compared to the ring-fixed flavone compounds, dibenzoylmethane intermediate is actually more flexible to be accommodated by glycosyltransferases or other decoration enzymes. Since C-C bonds is rigidly fixed in flavone skeleton, C-glycosylation on position C-6 and position C-8 of flavone is different and should be accomplished by two distinct CGTs (6-C-glycosyltransferase/8-C-glycosyltransferase, **Figure R4**). Such CGTs have been reported in some dicot plants [9,10], and their regio-specificity indicates that they are only specific to C-6 or C-8, but not both. CGTs recognizing 2-hydroxyflavones (tautomer: dibenzoylmethane), however, can complete C-glycosylation on C-6/C-8 at the same time due to the rotation of A ring in dibenzoylmethane (**Figure R4**). Many 6-C- and 8-C-isomer pairs (orientin/isorientin, vitexin/isovitexin, schaftoside/isoschaftoside) can be generated by the same CGTs. Therefore, in FNS I/FNS II-mediated pathway, at least three enzymes (FNS+6-C-glycosyltransferase+8-C-glycosyltransferase) are needed to produce both 6-C-glycosylated flavone and 8-C-glycosylated flavone, but by the mediation of dibenzoylmethane intermediate and taking advantage of non-enzymatic dehydration, two enzymes (F2H+6(8)-C-glycosyltransferase) are sufficient to accomplish both 6-C- and 8-C-glycosylation. Nature in fact efficiently use less decoration enzymes to generate more chemical diversity.

Figure R4. Nature's strategy to generate C-glycosylated flavones.

There are three ways to biosynthesize flavone skeleton (FNS I/FNS II/F2H-mediated pathway). Direct C-glycosylation on flavone skeleton involves regio-specific CGTs to produce 6-C- and 8-C-isomer, respectively. While through dibenzoylmethane intermediate, CGTs can accomplish 6-C- and 8-C-glycosylation at the same time. It seems that in the F2H-mediated pathway, less enzymes are needed to achieve the chemical diversity. Gly, glycosyl (glucosyl, arabinosyl, xylosyl.....); R indicates any substituents on the B ring.

(2) It would be better to combine the results from Fig3 and Fig4. The major point is the evolution of novel function for CGT gene family.

Response: We agree with the reviewer. In the previous manuscript, **Figure 3** and **Figure 4** both illustrated the results from **Result 2**. However, since reviewer #2 asked us to split **Result 2** (which contains too much information) into two parts, we have divided **Result 2** to two sub-sections: “**2. Comparative Genomics Reveals a Rich C-glycosyltransferase Reservoir**” and “**3. Characterization of C-glycosyltransferases and C-arabinosyltransferases**”. Therefore in the modified manuscript, **Figure 3** and **Figure 4** correspond to **Result 2** and **Result 3**, respectively. We fill it is better to keep these two figures separately in the revised manuscript.

Minor comments:

(1) Page 6. In Figure 2C, the scale bar beside the heatmap seems indicates the MS signal intensity, can authors quantify the CGF amount in different bamboos using absolute values? Since the authors had the standards, it would be better to provide the absolute quantification to compare the CGF accumulation in bamboo with other plants

Response: Thanks for the suggestion. We have quantified the absolute amounts of CGF compounds by standard curves (**Figure R5**). According to the MS signal intensity, the content of major CGF compounds in different *Phyllostachys* bamboos range from 0.3 mg/kg dry weight ~9972 mg/kg dry weight. We have modified the scale bar of **Figure 2C**.

Figure R5. Standard curves of C-glycosylated flavones generated by LC-MS.

Figure 2C. Heat map visualization indicates metabolic spectrum of C-monoglucosylated flavone in different tissues of five *Phyllostachys* bamboo. The content of major CGF compounds range from 0.3 mg/kg dry weight (DW) ~9972 mg/kg DW. L: leaf; S: stem; R: root; Sh: Shoot. Monoglucosides are highly accumulated in the leaves of *P. meyeri* McClure.

(2) In the second result, the authors described more than 30 C-glycosyltransferases (some have already been systematically named) from the grass family, I suggest the authors to name these genes/proteins through UGT Nomenclature Committee. Current numbering of these enzymes like *OsCGT1, 2, 3* is confusing.

Response: Thanks for the suggestion. We have named all the CGTs found in this research by submitting sequences to the UGT Nomenclature Committee. The formal names of grass family CGTs are listed in **Table R1**. UGT708A1~A8 were formally named by others. The newly added UGT708As are indicated in bold font. We have updated the information in **Table S2** in SI and also replaced the names like *OsCGT1,2,3...* with the formal names in the manuscript.

Table R1. Name of UGT708A suggested by UGT Nomenclature Committee

Formal Name	Name	Origin	Common name
UGT708A3	OsCGT	Oryza sativa japonica	Rice
UGT708A4	OsCGT2	Oryza sativa japonica	Rice
UGT708A2	OsCGT3	Oryza sativa japonica	Rice
UGT708A1	OsCGT4	Oryza sativa indica	Rice, long-grained
UGT708A39	OsCGT5	Oryza sativa indica	Rice, long-grained
UGT708A40	OsCGT6	Oryza sativa indica	Rice, long-grained
UGT708A6	ZmCGT	Zea mays L.	Maize
UGT708A5	ZmCGT2	Zea mays L.	Maize

UGT708A11	ZmCGT3	Zea mays L.	Maize
UGT708A41	ZmCGT4	Zea mays L.	Maize
UGT708A42	ZmCGT5	Zea mays L.	Maize
UGT708A14	TaCGT1-A	Triticum aestivum	Wheat
UGT708A52	TaCGT1-B	Triticum aestivum	Wheat
UGT708A53	TaCGT1-D	Triticum aestivum	Wheat
UGT708A15	TaCGT2-A	Triticum aestivum	Wheat
UGT708A54	TaCGT2-B	Triticum aestivum	Wheat
UGT708A55	TaCGT2-D	Triticum aestivum	Wheat
UGT708A7	BdCGT1	Brachypodium distachyon	Purple false brome
UGT708A8	BdCGT2	Brachypodium distachyon	Purple false brome
UGT708A36	SbCGT1	Sorghum bicolor (L.) Moench	Sorghum
UGT708A35	SbCGT2	Sorghum bicolor (L.) Moench	Sorghum
UGT708A34	SbCGT3	Sorghum bicolor (L.) Moench	Sorghum
UGT708A38	SbCGT4	Sorghum bicolor (L.) Moench	Sorghum
UGT708A31	SiCGT1	Setaria italica (L.) P. Beauvois	Foxtail millet
UGT708A32	SiCGT2	Setaria italica (L.) P. Beauvois	Foxtail millet
UGT708A33	SiCGT3	Setaria italica (L.) P. Beauvois	Foxtail millet
UGT708A43	PhCGT1	Phyllostachys edulis	Moso bamboo
UGT708A44	PgCGT1	Phyllostachys glauca McClure	/
UGT708A45	PpCGT1	Phyllostachys prominens W. Y. Xiong	/
UGT708A46	PhCGT2	Phyllostachys edulis	Moso bamboo
UGT708A48	PhCGT3	Phyllostachys edulis	Moso bamboo
UGT708A50	PhCGT4	Phyllostachys edulis	Moso bamboo

(3) Page 11. Line 235. “In order to chieve a high-yield”, I think it’s a mistake.

Response: Thanks for pointing out this issue. We have corrected this mistake in the manuscript.

(4) In the results of heterologous production of CGFs in *E. coli* systems, the authors did not mentioned the reason of choosing this host. Previous work in yeast (García Vanegas K, *Microbial Cell Factory* (2018) 17: 107) reported comparative yield of these compounds, which raised question of advantage of adopting *E. coli* as alternative host.

Response: Recent advance in the field of developing *E. coli* system as heterologous host for flavonoid production suggests that *E. coli* is not only suitable for the biosynthesis of flavonoid aglycones, but also flavonoid glycosides. By metabolic engineering, several successful case

reported high titer of diverse flavonoid glycosides. For example, 1.178 g of fisetin 3-*O*-glucoside and 1.026 g of fisetin 3-*O*-rhamnoside were produced in a 3 L fermenter/48 h while supplementing 0.9 g of fisetin [17]. In an UDP-xylose-supplied *E. coli* strain, 65.0 mg/L of quercetin 3-*O*-glucosyl (1→2) xyloside and 119.8 mg/L of rutin was produced [18]. Luteolin-7-*O*-glucuronide, quercetin-3-*O*-glucuronide and quercetin 3-*O*-galactoside could be biosynthesized to levels of 300, 687 and 280 mg/L when UDP-glucuronic acid/UDP-galactose biosynthesis were engineered [19]. *E. coli* is therefore a suitable microbial platform for the sustainable production of diverse glycosylated flavonoids.

(5) Figure 5B, in the illustration of cz261, the arrow and bar are misplaced.

Response: We apologize for the mistake in **Figure 5B**. The promoter and terminator were wrongly drawn. We have revised the marks in **Figure 5B** (also see response to reviewer #2).

Other minor revisions:

- (1) We have unified the binomial name of moso bamboo to “*Phyllostachys heterocycla*”. Although the name of “*Phyllostachys edulis*” is more widely accepted, “*P. heterocycla*” was used when the genome was sequenced in 2013 [20]. For this reason, many genes and genomic scaffolds were named with prefix “Ph”. We think it is better to keep this prefix in accordance with the previous recorded ones.
- (2) Page 5, Line 89, we deleted “in vitro assays”, it was a mistake.
- (3) According to the systematic naming by UGT Nomenclature Committee, we have modified the gene names in **Figure 3B**.

Reference

1. Hunskaar S, Hole K. The formalin test in mice: dissociation between inflammatory and non-inflammatory pain. *Pain* **30**, 103-114 (1987).
2. Garcia MM, Goicoechea C, Avellanal M, Traseira S, Martín MI, Sánchez-Robles EM. Comparison of the antinociceptive profiles of morphine and oxycodone in two models of inflammatory and osteoarthritic pain in rat. *Eur. J. Pharmacol.* **854**, 109-118 (2019).
3. Zhang Y, *et al.* A novel analgesic isolated from a Traditional Chinese Medicine. *Curr. Biol.* **24**, 117-123 (2014).
4. Farkhondeh T, Samarghandian S, Azimin-Nezhad M, Samini F. Effect of chrysin on nociception in formalin test and serum levels of noradrenalin and corticosterone in rats. *Int. J. Clin. Exp. Med.* **8**, 2465-2470 (2015).
5. Chen DW, *et al.* Probing the catalytic promiscuity of a regio- and stereospecific C-glycosyltransferase from *Mangifera indica*. *Angew. Chem. Int. Edit.* **54**, 12678-12682 (2015).
6. Nagatomo Y, Usui S, Ito T, Kato A, Shimosaka M, Taguchi G. Purification, molecular cloning and functional characterization of flavonoid C-glycosyltransferases from *Fagopyrum esculentum* M. (buckwheat) cotyledon. *Plant J.* **80**, 437-448 (2014).
7. Hirade Y, Kotoku N, Terasaka K, Saijo-Hamano Y, Fukumoto A, Mizukami H. Identification and functional analysis of 2-hydroxyflavanone C-glycosyltransferase in soybean (*Glycine max*). *FEBS Lett.* **589**, 1778-1786 (2015).
8. Ito T, Fujimoto S, Suito F, Shimosaka M, Taguchi G. C-Glycosyltransferases catalyzing the formation of di-C-glucosyl flavonoids in citrus plants. *Plant J.* **91**, 187-198 (2017).
9. Sasaki N, *et al.* Identification of the glucosyltransferase that mediates direct flavone C-glucosylation in *Gentiana triflora*. *FEBS Lett.* **589**, 182-187 (2015).
10. He J-B, *et al.* Molecular and structural characterization of a promiscuous C-glycosyltransferase from *Trollius chinensis*. *Angew. Chem. Int. Edit.* **131**, 11637-11644 (2019).
11. Brazier-Hicks M, Evans KM, Gershater MC, Puschmann H, Steel PG, Edwards R. The C-glycosylation of flavonoids in cereals. *J. Biol. Chem.* **284**, 17926-17934 (2009).
12. Wang X, Li C, Zhou C, Li J, Zhang Y. Molecular characterization of the C-glucosylation for puerarin biosynthesis in *Pueraria lobata*. *Plant J.* **90**, 535-546 (2017).
13. Du Y, Chu H, Chu IK, Lo C. CYP93G2 is a flavanone 2-hydroxylase required for C-

- glycosylflavone biosynthesis in rice. *Plant Physiol.* **154**, 324-333 (2010).
14. Morohashi, K. *et al.* A genome-wide regulatory framework identifies maize *Pericarp Color1* controlled genes. *Plant Cell* **24**, 2745, (2012).
 15. Akashi, T., Aoki, T. & Ayabe, S.-i. Identification of a cytochrome P450 cDNA encoding (2S)-flavanone 2-hydroxylase of licorice (*Glycyrrhiza echinata* L.; Fabaceae) which represents licodione synthase and flavone synthase II 1. *FEBS Lett.* **431**, 287-290 (1998).
 16. Zhang, J., Subramanian, S., Zhang, Y. & Yu, O. Flavone synthases from *Medicago truncatula* are flavanone-2-hydroxylases and are important for nodulation. *Plant Physiol.* **144**, 741-751, doi:10.1104/pp.106.095018 (2007).
 17. Parajuli, P., Pandey, R. P., Trang, N. T. H., Chaudhary, A. K. & Sohng, J. K. Synthetic sugar cassettes for the efficient production of flavonol glycosides in *Escherichia coli*. *Microb. Cell Fact.* **14**, 76 (2015).
 18. An, D. G., Yang, S. M., Kim, B. G. & Ahn, J.-H. Biosynthesis of two quercetin *O*-diglycosides in *Escherichia coli*. *J. Ind. Microbiol. Biotechnol.* **43**, 841-849 (2016).
 19. Kim, S. Y., Lee, H. R., Park, K.-s., Kim, B.-G. & Ahn, J.-H. Metabolic engineering of *Escherichia coli* for the biosynthesis of flavonoid-*O*-glucuronides and flavonoid-*O*-galactoside. *Appl. Microbiol. Biotechnol.* **99**, 2233-2242 (2015).
 20. Peng Z, *et al.* The draft genome of the fast-growing non-timber forest species moso bamboo (*Phyllostachys heterocycla*). *Nat. Genet.* **45**, 456-461 (2013).

REVIEWERS' COMMENTS:

Reviewer #1 (Remarks to the Author):

The manuscript was revised and the questions were answered with satisfactory. Some necessary information were added to support the conclusion. Some minor errors were corrected in this revised manuscript. The work confirmed bamboo species as a rich source of C-glycosylated flavonoids (CGFs) with neuro-analgesic and anti-inflammatory effects, and made clear the biosynthetic pathway, which is novel and meaningful. I suggest that the manuscript be accepted.

Reviewer #2 (Remarks to the Author):

The manuscript of Pathway-specific Enzymes from Bamboo and Crop Leaves Biosynthesize Antinociceptive C-glycosylated Flavones has been significantly improved compared to the earlier version. Most of my concerns have been well addressed. However, some issues remain which should be taken care of before publication.

1. Please confirm the structures of the C-glycosylated products and the aglycons shown in Fig 4C and 4E. Generally, there are two different structures for 2-hydroxynaringenin (ring-open and ring-close).
2. As the CGTs of Clade B were assigned as C-arabinosyltransferases, the comparison of the Km values of UDP-Glc and UDP-Ara should be provided. The authors only provided the Km values of acceptors which cannot strengthen their conclusions. Alternatively, the authors should revise their statement in this part to make it more precise.

Reviewer #3 (Remarks to the Author):

All my recommendations and comments have been acted properly upon. I have no further requirements for the authors.

Response to Reviewer Comments (COMMSBIO-19-0957B)

Thank you very much for considering publication of our manuscript COMMSBIO-19-0957B. We thank the reviewers again for their constructive and positive comments regarding this manuscript. Since Reviewer #1 and #3 did not have additional concerns, only response to Reviewer #2 is addressed here. We also revised and reformatted the manuscript and Supplementary Information according to the editorial requests. The changes made in the manuscript and SI are highlighted.

Reviewer #2 (Remarks to the Author):

The manuscript of *Pathway-specific Enzymes from Bamboo and Crop Leaves Biosynthesize Antinociceptive C-glycosylated Flavones* has been significantly improved compared to the earlier version. Most of my concerns have been well addressed. However, some issues remain which should be taken care of before publication.

1. Please confirm the structures of the C-glycosylated products and the aglycones shown in Fig 4C and 4E. Generally, there are two different structures for 2-hydroxynaringenin (ring-open and ring-close).

Response: According to the report of C-glycosylated flavone biosynthesis thus far, 2-hydroxynaringenin (2OHNar) has been considered as a key intermediate. All the existing cases mentioned that the close circular form of 2OHNar was a “metastable” compound, which existed in equilibrium with its open-circular dibenzoylmethane form in the solvent¹⁻⁵. It is hard to distinguish which form is the real conformation when C-glycosylation occurs, as both forms have been proposed as substrates of CGTs.

Most reports supported the open-ring form as the actual substrate [Fig. 1B in ref 1, Fig.1(a) in ref 2, Fig. 1 in ref 3], due to: (1) The open chain form facilitated electrophilic aromatic substitution by UDP-sugar donors; (2) The open chain form was more flexible to be accommodated by corresponding C-glycosyltransferases. Many reports also pointed out that

substrates with B ring-fixed structures (like flavanone, flavone and flavonol) were generally not accepted by the CGTs¹⁻³. In the case of MiCGT from mango, benzophenone intermediate (maclurin, which could be regarded as a “B ring-opened” xanthone) was identified as the intermediate in the biosynthetic pathway to mangiferin, while the ring-fixed xanthone substrate (like norathyriol) was not C-glycosylated⁶. Hitherto, only ZmUGT708A6 from maize has been reported to act on the close-ring form of 2-hydroxyflavanone, because less structural flexibility is necessary to generate a biased flavone 6-C-glucoside formation⁴.

In our case, we proposed the open-ring form of 2OHNar as suitable substrate for C-glycosylation for two reasons: (1) Being consistent with OsCGT/FeCGT/FcCGT/CuCGT¹⁻³, CGTs found in this work are not active on the ring-fixed flavanone (i.e, naringenin) and flavone (i.e, apigenin); (2) Our *in vivo* test in *E. coli* suggested a ratio of 6:4 of flavone 8-C-glucoside to flavone 6-C-glucoside. Predominant production of flavone 6-C-glucoside were not observed like that was reported in maize. Therefore we revised the description the C-glycosylated products and the aglycones in Fig. 4c,e.

2. As the CGTs of Clade B were assigned as C-arabinoxyltransferases, the comparison of the Km values of UDP-Glc and UDP-Ara should be provided. The authors only provided the Km values of acceptors which cannot strengthen their conclusions. Alternatively, the authors should revise their statement in this part to make it more precise.

Response: We agree with the reviewer. We have revised the description in result 3 and Fig. 4d, e. Clade B was identified as CGTs recognizing both UDP-glucose and UDP-arabinose. Therefore we avoided designating them as C-arabinoxyltransferases which might cause misconception. It is difficult to measure the K_m of all the twelve Clade B CGTs, since UDP-arabinose is not easily accessible.

Revisions according to the editorial requests:

- (1) We have shortened the abstract. Now it is 144 words.
- (2) We have added the section title “Introduction” and removed numbers from the other subsection titles.
- (3) We have renamed supplementary items and figures.

- (4) We replaced or removed “novel” in the text.
- (5) We indicated the dilution ratio of antibodies used in Methods and Materials.
- (6) We removed “financial” in the COI statement.
- (7) We added DAS, author contributions, and acknowledgements in the text
- (8) We converted the bar graphs in Fig.5, Fig. 6 and Supplementary Fig. 15,16,18,19.
- (9) We moved Supplementary Fig. 2 to Supplementary Table 1, and reordered the supplementary items.
- (10) We deleted the cover page and table of content in SI.

Reference

1. Brazier-Hicks M, Evans KM, Gershter MC, Puschmann H, Steel PG, Edwards R. The C-glycosylation of flavonoids in cereals. *J. Biol. Chem.* **284**, 17926-17934 (2009).
2. Nagatomo Y, Usui S, Ito T, Kato A, Shimosaka M, Taguchi G. Purification, molecular cloning and functional characterization of flavonoid C-glucosyltransferases from *Fagopyrum esculentum* M. (buckwheat) cotyledon. *Plant J.* **80**, 437-448 (2014).
3. Ito T, Fujimoto S, Suito F, Shimosaka M, Taguchi G. C-glycosyltransferases catalyzing the formation of di-C-glucosyl flavonoids in citrus plants. *Plant J.* **91**, 187-198 (2017).
4. Ferreyra MLF, Rodriguez E, Casas MI, Labadie G, Grotewold E, Casati P. Identification of a bifunctional maize C- and O-glucosyltransferase. *J. Biol. Chem.* **288**, 31678-31688 (2013).
5. Hirade Y, Kotoku N, Terasaka K, Saijo-Hamano Y, Fukumoto A, Mizukami H. Identification and functional analysis of 2-hydroxyflavanone C-glucosyltransferase in soybean (*Glycine max*). *FEBS Lett.* **589**, 1778-1786 (2015).
6. Chen DW, *et al.* Probing the catalytic promiscuity of a regio- and stereospecific C-glycosyltransferase from *Mangifera indica*. *Angew. Chem. Int. Edit.* **54**, 12678-12682 (2015).